# Direct control of lysosomal catabolic activity by mTORC1 through regulation of V-ATPase assembly

Edoardo Ratto [1,2], S. Roy Chowdhury[1], Nora S. Siefert [1], Martin Schneider[3], Marten Wittmann[1], Dominic Helm [3] & Wilhelm Palm [1] ✉

Mammalian cells can acquire exogenous amino acids through endocytosis and lysosomal catabolism of extracellular proteins. In amino acid-replete environments, nutritional utilization of extracellular proteins is suppressed by the amino acid sensor mechanistic target of rapamycin complex 1 (mTORC1) through an unknown process. Here, we show that mTORC1 blocks lysosomal degradation of extracellular proteins by suppressing V-ATPase-mediated acidification of lysosomes. When mTORC1 is active, peripheral V-ATPase $V_1$ domains reside in the cytosol where they are stabilized by association with the chaperonin TRiC. Consequently, most lysosomes display low catabolic activity. When mTORC1 activity declines, V-ATPase $V_1$ domains move to membrane-integral V-ATPase $V_o$ domains at lysosomes to assemble active proton pumps. The resulting drop in luminal pH increases protease activity and degradation of protein contents throughout the lysosomal population. These results uncover a principle by which cells rapidly respond to changes in their nutrient environment by mobilizing the latent catabolic capacity of lysosomes.

The cells which compose a mammalian organism reside in complex nutrient environments that provide diverse metabolites and macromolecular nutrients[1]. Amino acids are vital cellular nutrients, serving as substrates for various metabolic pathways and constituting the building blocks of proteins. Mammalian cells readily acquire exogenous amino acids in their monomeric form through plasma membrane transporter-mediated uptake. However, the vast majority of amino acids in circulation and in pericellular fluids is contained within proteins[2]. When facing shortages in monomeric amino acids, cells can utilize the copious, non-dedicated nutrient stores of extracellular proteins to sustain cellular functions.

Cells unlock the amino acid contents of extracellular proteins in the lysosome, a degradative organelle whose lumen contains various proteases and other hydrolytic enzymes[3]. The acidic environment in which lysosomal enzymes are active is generated through the vacuolar-type H⁺ ATPase (V-ATPase), a proton pump that consists of

a peripheral $V_1$ domain, which hydrolyses ATP, and a membrane-integral $V_o$ domain, which translocates protons into the lysosomal lumen[4,5]. Proteins are supplied from the extracellular space to the lysosome through endocytosis. In particular, the non-selective endocytic pathway of macropinocytosis enables cells to ingest extracellular proteins in sufficient quantities to sustain viability and growth during amino acid starvation. Macropinocytosis is commonly upregulated in cancer, which allows malignant cells to grow in poorly vascularized, nutrient-poor tumour microenvironments[6–8]. The lysosome also catabolizes proteins from intracellular sources, which are supplied by macroautophagy, hereafter referred to as autophagy. By recycling the nutritional content of intracellular proteins, autophagy can sustain cell survival during transient periods of starvation[9].

Cells employ nutrient-sensing signalling pathways to constantly monitor intracellular metabolite levels and adjust metabolic activities

[1]Cell Signaling and Metabolism, German Cancer Research Center (DKFZ), Heidelberg, Germany. [2]Faculty of Biosciences, University of Heidelberg, Heidelberg, Germany. [3]MS-based Protein Analysis Unit, Genomics and Proteomics Core Facility, German Cancer Research Center (DKFZ), Heidelberg, Germany. ✉e-mail: w.palm@dkfz-heidelberg.de

accordingly. The serine/threonine kinase mechanistic target of rapamycin complex 1 (mTORC1) is the central cellular amino acid sensor[10,11]. When amino acids are abundant, mTORC1 moves to lysosomal membranes where the kinase subsequently becomes activated[12]. In turn, mTORC1 regulates multiple effector proteins that concertedly suppress protein catabolism while stimulating protein synthesis. Catabolism of intracellular proteins is blocked by mTORC1 via two main pathways: mTORC1 prevents autophagosome formation by inhibiting the autophagy initiator kinases Unc51-like kinase 1/2 (Ulk1/2)[13], and suppresses expression of lysosomal and autophagic genes by inducing cytoplasmic retention of Transcription Factor EB (TFEB)[14,15]. Notably, although mTORC1 activation is orchestrated at lysosomal membranes, both these pathways regulate lysosomal function indirectly—cytosolic formation of autophagosomes and changes in nuclear gene expression. mTORC1 also inhibits lysosomal catabolism of proteins that were taken up from the environment[8]. However, the mechanism through which mTORC1 prevents nutrient generation from extracellular proteins has been unknown.

Here, we show that mTORC1 suppresses the nutritional use of extracellular proteins by directly controlling the catabolic activity of lysosomes. mTORC1 inactivation triggers assembly of the V-ATPase at lysosomal membranes, which acidifies the organelle lumen and activates lysosomal proteases. By increasing catabolic activity throughout the lysosomal population, cells initiate degradation of protein contents from extra- and intracellular sources that were accumulated in lysosomes. Thus, by mobilizing the latent catabolic capacity of lysosomes, mTORC1 inactivation enables rapid adaptation of cells to changes in nutrient supply.

## Results

### mTORC1 blocks lysosomal catabolism of endocytosed proteins

mTORC1 blocks intracellular nutrient generation from proteins that were taken up from the environment[8]. To identify the underlying mechanism, we first asked whether mTORC1 suppressed endocytic uptake or lysosomal catabolism. Cells constantly engage in non-selective uptake of extracellular proteins through fluid-phase endocytosis and can internalize large quantities of proteins through macropinocytosis[1]. To probe for these endocytic pathways, SV40 large T antigen-immortalized mouse embryonic fibroblasts (MEFs) were fed fluorescently labelled 10 kDa dextran, a general marker for fluid-phase endocytosis, or 70 kDa dextran, which enters cells through macropinocytosis[16]. Inhibiting mTORC1 signalling with torin 1 did not alter uptake of 10 kDa dextran (Fig. 1a, b) or 70 kDa dextran (Fig. 1c, d). Thus, mTORC1 does not suppress constitutive fluid-phase endocytosis or macropinocytosis. Next, we investigated whether mTORC1 regulated lysosomal proteolysis using DQ BSA, a self-quenched albumin probe that fluoresces upon degradation[17]. To monitor lysosomal catabolism directly, without confounding contribution of upstream endocytic events, lysosomes were loaded with DQ BSA for 4 to 6 h followed by 3 h chase in fresh medium (Fig. 1e). After the chase period, intracellular DQ BSA fluorescence did not increase, indicating that no further lysosomal accumulation or degradation of DQ BSA took place (Supplementary Fig. 1a). However, treating cells with DQ BSA-loaded lysosomes after the chase period with torin 1 caused a strong fluorescence increase within 1 h (Fig. 1f, g). Thus, mTORC1 inhibition rapidly triggers degradation of extracellular proteins that were accumulated in lysosomes through endocytosis.

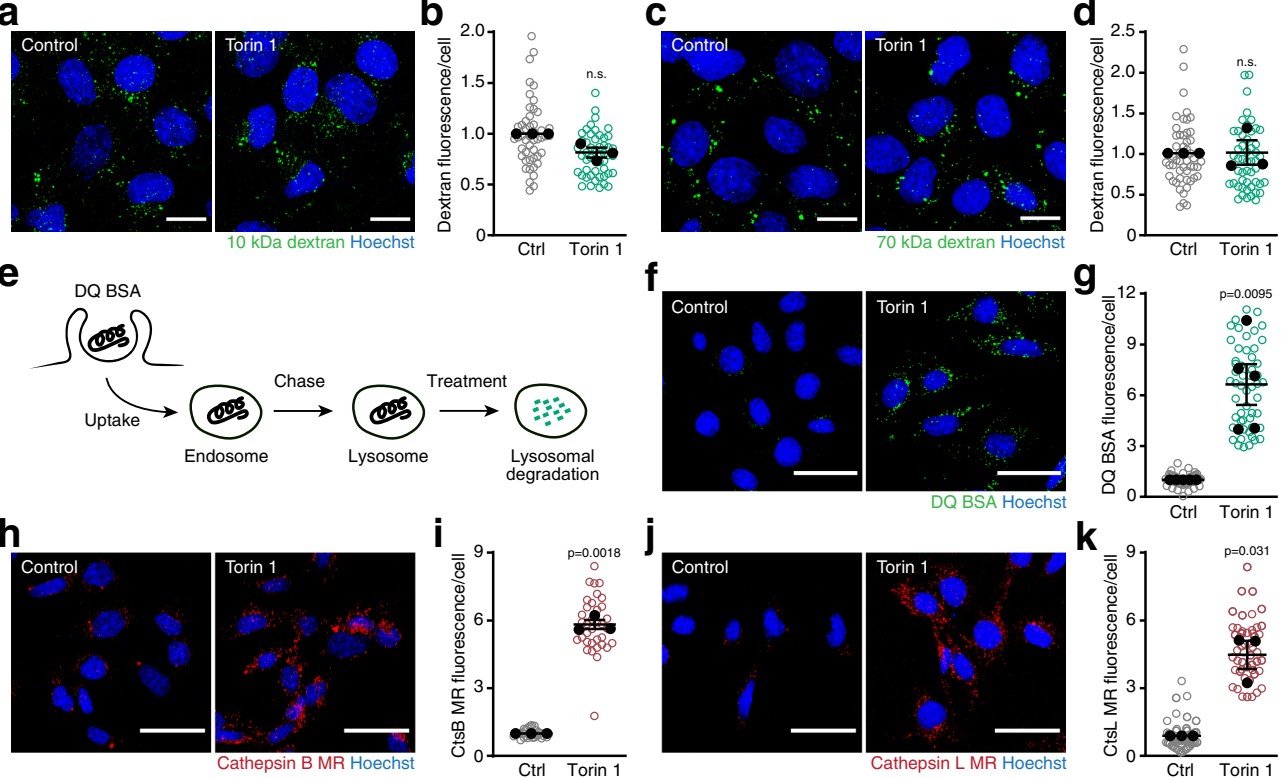

**Fig. 1 | Degradation of extracellular proteins is suppressed by mTORC1 at the step of lysosomal catabolism. a–d** Intracellular levels of fluorescently labelled **a, b** 10 kDa dextran, **c, d** 70 kDa dextran in MEFs after 45 min of uptake. Torin 1 [400 nM] was added 15 min before the experiment. Scale bars = 20 μm.
**e** Schematic of lysosomal protein degradation assay based on preloading lysosomes with DQ BSA and measuring dequenching of DQ BSA fluorescence upon proteolysis. **f, g** Degradation of lysosomally preloaded DQ BSA in MEFs after

1 h ± torin 1 [400 nM]. Scale bars = 50 μm. **h–k** Degradation of magic red (MR) substrates for **h, i** cathepsin B, **j, k** cathepsin L in MEFs after 1 h ± torin 1 [250 nM]. Scale bars = 50 μm. Data are normalized replicate mean ± SEM (closed circles) and fields of view (open circles; ≥10 per replicate). **b, d, i, k** $n = 3$, **g** $n = 5$ biologically independent experiments. p-values were calculated using a two-tailed unpaired t-test with Welch correction. n.s. not significant. Source data are provided as a Source data file.

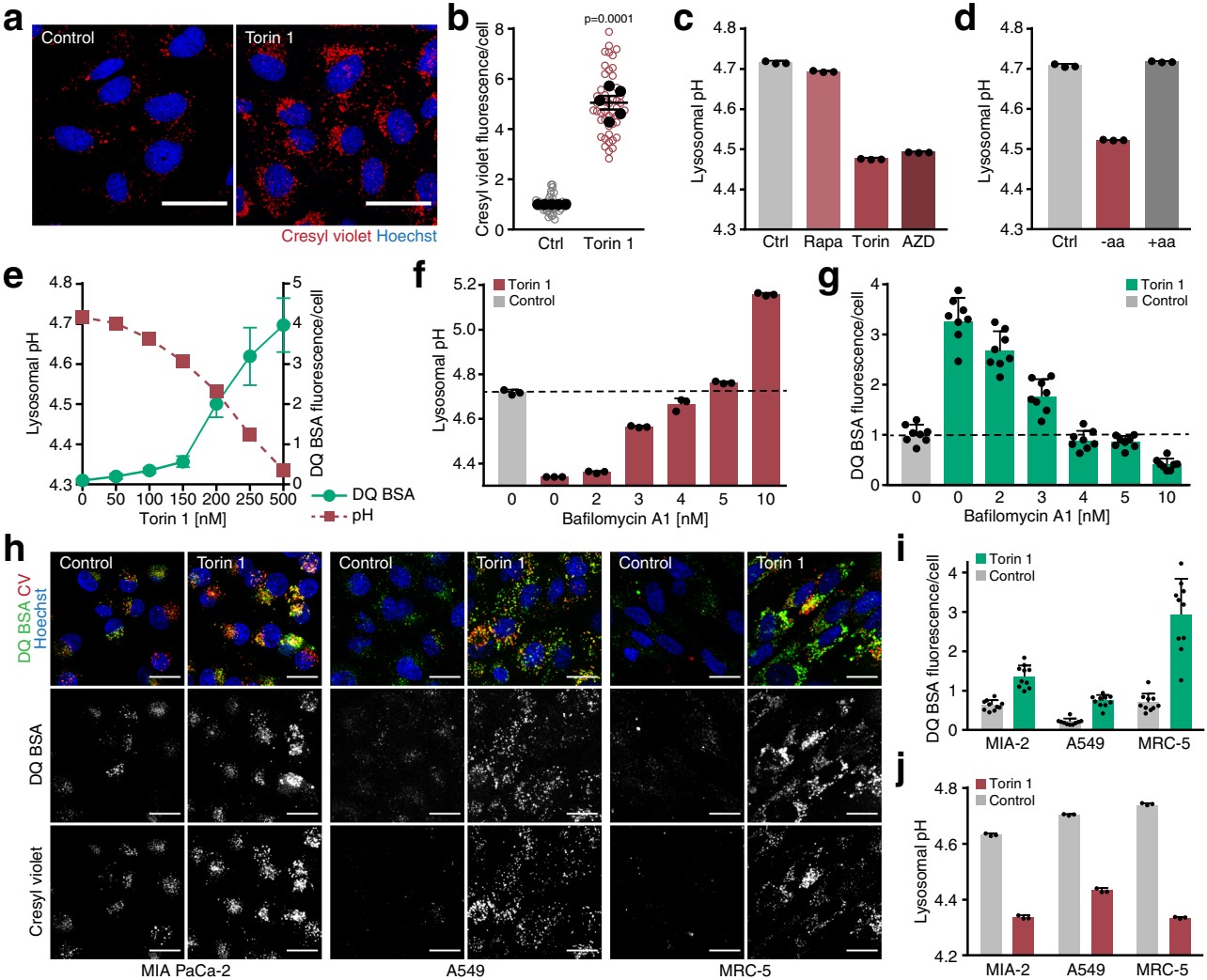

**Fig. 2 | mTORC1 inactivation increases lysosomal protein catabolism through lysosomal acidification. a** Cresyl violet accumulation in MEFs after 1 h ± torin 1 [400 nM]. Scale bars = 50 μm. **b** Quantification of cresyl violet accumulation in cells treated as in **a**; data are normalized replicate mean ± SEM (closed circles) and fields of view (open circles; ≥8 per replicate). **c** Lysosomal pH of MEFs after 1 h rapamycin [100 nM], torin 1 [250 nM], or AZD8055 [250 nM], quantified by lysosensor. **d** Lysosomal pH of MEFs after 1 h amino acid starvation (−aa), or 1 h aa starvation + 30 min aa restimulation (+aa), quantified by lysosensor. **e** Lysosomal pH, quantified by lysosensor, and degradation of lysosomally preloaded DQ BSA in MEFs after 1 h torin 1 at indicated concentrations. Measurements were conducted separately under identical conditions. **f** Lysosomal pH, quantified by lysosensor, in MEFs after 1 h torin 1 [400 nM] and bafilomycin A1 at indicated concentrations. The dashed line indicates mean lysosomal pH of control cells. **g** Degradation of lysosomally preloaded DQ BSA in MEFs after 1 h torin 1 [400 nM] and bafilomycin A1 at indicated concentrations. The dashed line indicates mean DQ BSA fluorescence of control cells. **h** Degradation of lysosomally preloaded DQ BSA and accumulation of cresyl violet (CV) in human carcinoma cell lines (MIA PaCa-2, A549) and fibroblasts (MRC-5) after 3 h ± torin 1 [400 nM]. Scale bars = 20 μm. **i** Quantification of DQ BSA fluorescence of cell lines shown in **h**. **j** Lysosomal pH in cell lines as in **h** after 1 h ± torin 1, quantified by lysosensor. **b** n = 5 biologically independent experiments; p-value was calculated using a two-tailed unpaired t-test with Welch correction. **c**–**f**, **j** Lysosensor data are mean ± SD (3 technical replicates). **e**, **g**, **i** DQ BSA data are mean ± SD (8–10 fields of view). **c**–**j** One representative of n = 3 biologically independent experiments. Source data are provided as a Source data file.

Conceivably, the increase in lysosomal albumin catabolism in response to mTORC1 inhibition could result from a change in the abundance or activity of lysosomal proteases. Torin 1 treatment did not alter the abundance of mature cathepsins (Supplementary Fig. 1b). Thus, we examined whether mTORC1 inhibition regulated lysosomal protease activity. To this end, we used cell-permeable Magic Red (MR) substrates for the ubiquitous lysosomal proteases cathepsin B and cathepsin L, whose fluorescence becomes de-quenched upon hydrolysis[18]. Cells were pretreated for 1 h with torin 1, and then briefly incubated with cathepsin MR substrates. Torin 1 caused a strong increase in the fluorescence of cathepsin B MR (Fig. 1h, i) and cathepsin L MR (Fig. 1j, k). To test whether specific lysosomal proteases were required for the increase in protein degradation, we genetically ablated the abundant cathepsins B, D or L using CRISPR/Cas9 (Supplementary Fig. 1c). Loss of each of these cathepsins individually did not

impair the high levels of DQ BSA degradation in torin 1-treated cells (Supplementary Fig. 1d, e). By contrast, pharmacological inhibition of all lysosomal proteases strongly suppressed DQ BSA degradation. Thus, mTORC1 inhibition triggers lysosomal protein degradation through a global increase in lysosomal protease activity.

One process through which mTORC1 inactivation promotes lysosomal catabolism is the induction of lysosomal gene expression[14,15]. However, torin 1 strongly increased DQ BSA degradation even when protein synthesis was blocked with the translation inhibitor cycloheximide (Supplementary Fig. 1f, g). mTORC1 inactivation also induces the formation of autophagosomes, which supply cytosolic constituents to lysosomal catabolism[9]. Genetic ablation of autophagy through deletion of Atg5 did not impair basal levels of lysosomal DQ BSA degradation or the increase in DQ BSA degradation in response to torin 1 (Supplementary Fig. 1h–j). Overall, these findings

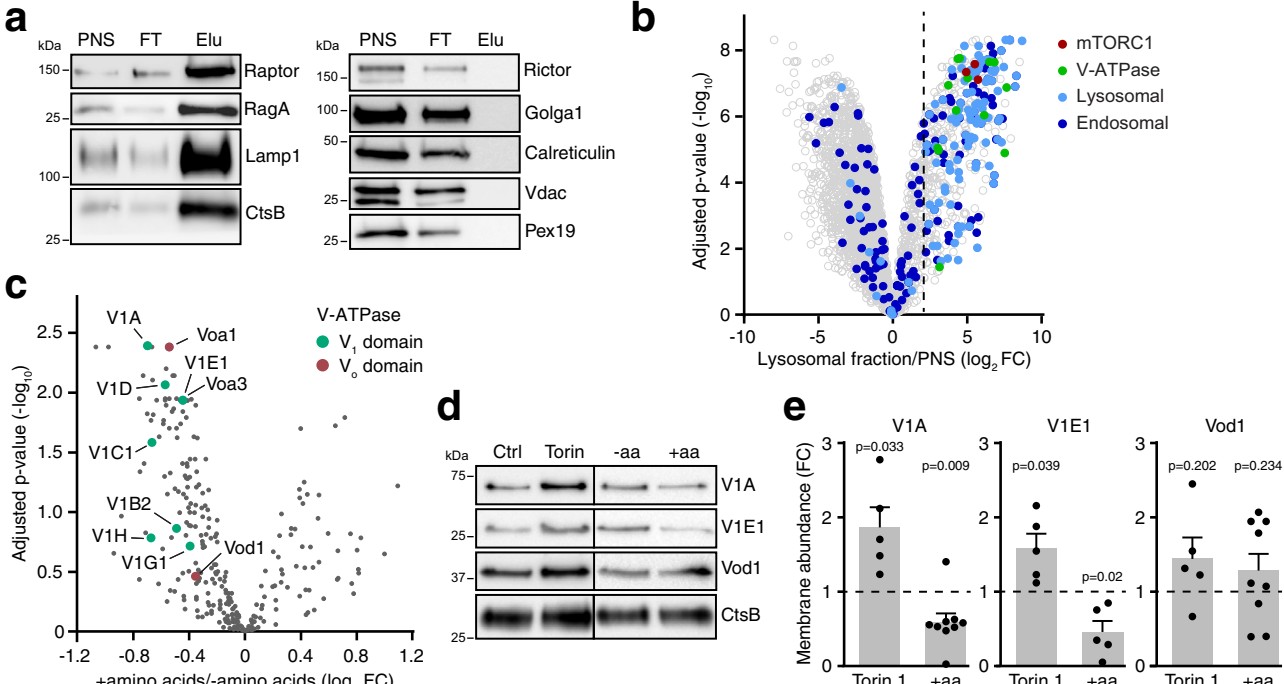

**Fig. 3 | Lysosomal V-ATPase increases in response to mTORC1 inactivation.**
**a** Magnetic enrichment of lysosomes from MEFs, analysed by western blot. PNS: post nuclear supernatant; FT: column flow through; Elu: lysosomal eluate. Contaminating organelle markers are Golga1 (Golgi), Calreticulin (ER), Vdac (mitochondria), Pex19 (peroxisomes). **b** Enriched proteins in lysosomal fractions, quantified by label-free mass spectrometry. The dashed line (log$_2$ fold change >2) demarcates the cut-off for lysosomal proteins. **c** Changes in lysosomal proteins after 1 h aa starvation + 30 min aa restimulation (+aa) versus 1 h aa starvation (−aa) in MEFs, quantified by label-free mass spectrometry. **d** V-ATPase subunits in membrane fractions of dextran-loaded lysosomes from MEFs after 1 h ± torin 1 [400 nM], or after 1 h amino acid starvation ± 30 min aa restimulation (±aa), analysed by western blot. **e** V-ATPase subunit abundance changes in membrane fractions, quantified by western blot for torin 1/control and +aa/−aa as in **d**. Data are normalized replicate mean ± SEM. Dashed lines indicate protein abundance in the control/−aa groups. **a** One representative of $n = 5$ biologically independent experiments. **b**, **c** $n = 5$, **e** $n = 5$ (V1A, V1E, Vod1 + torin 1, V1E + aa) or $n = 9$ (V1A, Vod1 +aa) biologically independent experiments. In **b**, **c**, adjusted $p$-values were calculated using limma moderated $t$-statistic and adjusted with the Benjamini–Hochberg method for multiple testing. In **e**, $p$-values were calculated using a one sample $t$-test with a hypothetical mean of 1. Source data are provided as a Source data file.

indicate that mTORC1 inhibition enhances lysosomal cathepsin activity and protein degradation through a mechanism that acts over short time scales directly at lysosomes.

## mTORC1 inactivation increases lysosomal proteolysis through lysosomal acidification

We noted that brief treatment of MEFs with torin 1 caused a strong increase in the acidotrophic fluorophores cresyl violet[19] and lysotracker red, which accumulate in lysosomes (Fig. 2a, b, Supplementary Fig. 2a, b). This raised the possibility that mTORC1 inactivation induced a change in lysosomal pH. To test this, lysosomes were loaded with dextran-conjugated lysosensor yellow/blue DND-160 (lysosensor), a ratiometric dye that allows absolute quantification of lysosomal pH (Supplementary Fig. 2c). Within 1 h of treatment with torin 1 or an additional mTOR kinase inhibitor, AZD8055, lysosomal pH decreased from pH 4.7 to pH 4.5 (Fig. 2c). However, lysosomal pH, cresyl violet accumulation and DQ BSA degradation were not affected by rapamycin (Fig. 2c, Supplementary Fig. 2d–g), which acts as a partial mTORC1 inhibitor (ref. 20, Supplementary Fig. 2h). To modulate mTORC1 activity by physiological means, cells were subjected to 1 h amino acid starvation or to 1 h amino acid starvation followed by 30 min amino acid restimulation (Supplementary Fig. 2i). Amino acid starvation decreased lysosomal pH to the same level as pharmacological inhibition of mTORC1 (Fig. 2d). Amino acid restimulation of starved cells increased lysosomal pH to its resting state within 30 min. Thus, lysosomal pH rapidly responds to changes in mTORC1 activity.

To further characterise the relationship of lysosomal protein catabolism and lysosomal pH, we examined the concentration-dependent effects of mTOR inhibitors. Raising torin 1 concentrations

from 50 to 500 nM progressively increased DQ BSA degradation, which correlated with a concomitant decrease in lysosomal pH (Fig. 2e), consistent with the acidic pH optimum of lysosomal proteases[21]. To confirm that the change in pH was responsible for the increased proteolytic activity, we sought to retain lysosomal pH at resting level in the context of mTORC1 inhibition. At 3–5 nM, the V-ATPase inhibitor bafilomycin A1 progressively abrogated the torin 1-induced drop in lysosomal pH (Fig. 2f). The torin 1-induced increase in lysosomal DQ BSA degradation was suppressed over the same concentration range (Fig. 2g). Consistently, low concentrations of bafilomycin A1 were sufficient to block the increase in cathepsin B activity in response to torin 1 (Supplementary Fig. 2j), and suppressed turnover of the autophagosomal protein LC3-II, which is degraded upon reaching the lysosome (Supplementary Fig. 2k). Of note, low concentrations of bafilomycin A1 did not perturb mTORC1 signalling (Supplementary Fig. 2l). Together, these results indicate that mTORC1 inactivation increases lysosomal proteolytic activity by lowering lysosomal pH.

To substantiate the generality of the above findings, we examined human carcinoma cell lines harbouring oncogenic KRAS mutations, which promote uptake of extracellular proteins through macropinocytosis[6], and immortalized human fibroblasts. Torin 1 caused a strong increase in degradation of lysosomally loaded DQ BSA in the different cell lines (Fig. 2h, i). Consistently, torin 1 caused lysosomal cresyl violet accumulation (Fig. 2h; Supplementary Fig. 2m), which correlated with a decrease in lysosomal pH from a resting state of pH 4.6–4.8 down to pH 4.3–4.4 in the different cell lines (Fig. 2j). Thus, various mammalian cell types respond to mTORC1 inactivation with a drop in lysosomal pH and resulting increase in lysosomal proteolysis.

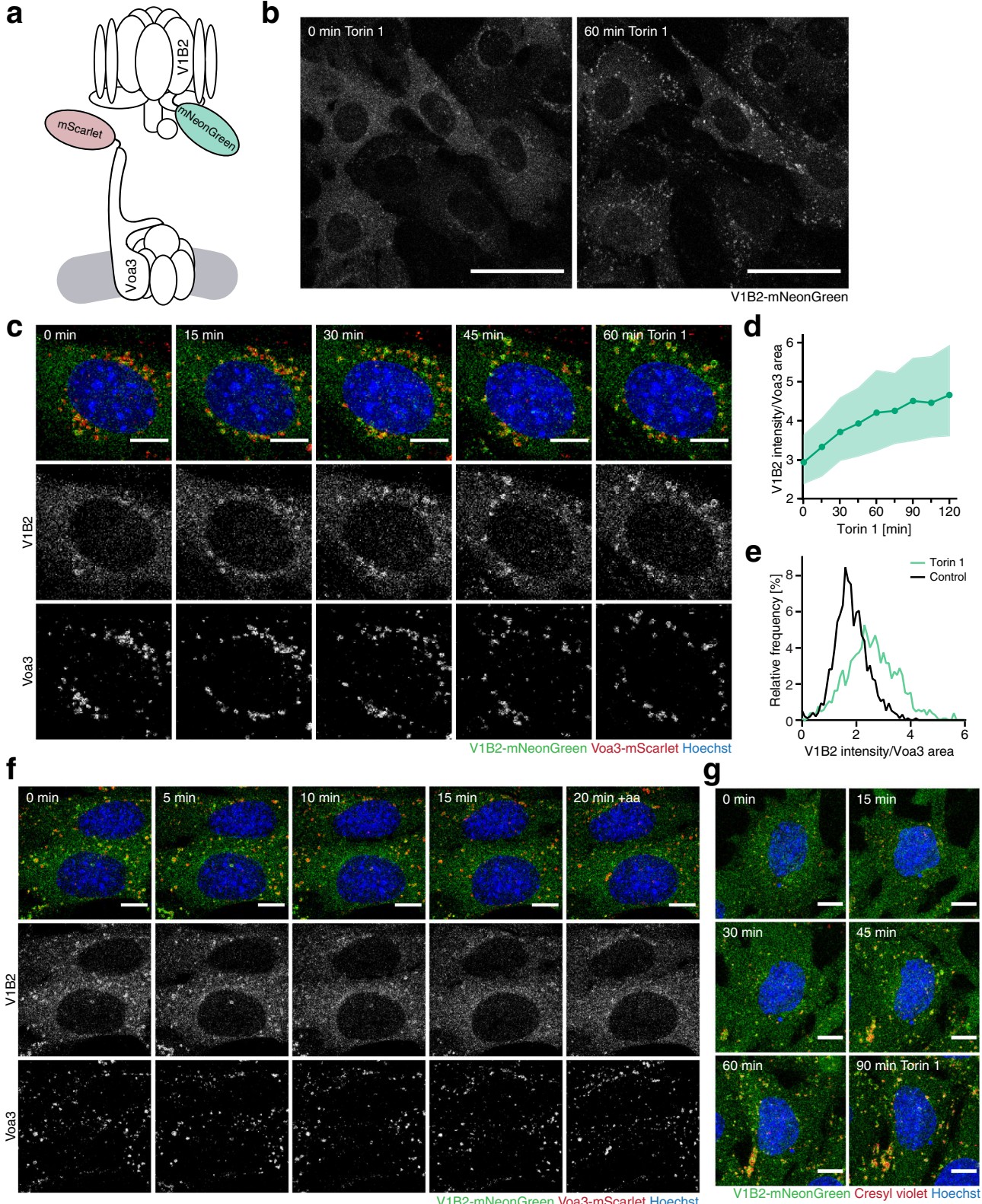

**Fig. 4 | mTORC1 controls reversible assembly of the V-ATPase at lysosomes.**
**a** Schematic of the live cell imaging assay for lysosomal V-ATPase assembly.
**b** Subcellular localization of V1B2-mNeonGreen before and 60 min after
treatment with torin 1 [400 nM]. Scale bars = 50 μm. **c** Subcellular localization
of V1B2-mNeonGreen and Voa3-mScarlet in MEFs over time after addition of
torin 1 [400 nM]. Scale bars = 10 μm. **d** Quantification of V1B2-mNeonGreen
recruitment to Voa3-mScarlet-containing lysosomes in cells shown in **c**. Data
are represented as median ± IQR (≥7000 organelles in 12 fields of view with a
total of ≥140 cells). **e** Relative intensity distribution of V1B2-mNeonGreen in
Voa3-mScarlet-containing lysosomes before and after 1 h torin 1 in cells
shown in **c**. **f** Subcellular localization of V1B2-mNeonGreen and Voa3-
mScarlet in MEFs over time after 1 h amino acid starvation followed by amino
acid restimulation (+aa). Scale bars = 10 μm. **g** Subcellular localization of
V1B2-mNeonGreen and cresyl violet in MEFs over time upon torin 1 treatment
[400 nM]. Scale bars = 10 μm. One representative of **b**, **g** $n = 5$, **c**–**f** $n = 3$ bio-
logically independent experiments.

## The V-ATPase assembles at lysosomes in response to mTORC1 inactivation

To identify the mechanism through which mTORC1 regulates lysosomal pH, we quantified mTORC1-responsive changes in lysosomal proteins. To this end, we first generated an organelle-specific proteome by loading the endolysosomal compartment of MEFs with dextran-coated magnetic nanoparticles, DexoMAG, for subsequent enrichment over magnetic columns[22]. After 14 h incubation, most DexoMAG localized to lysosomes that responded to torin 1 treatment with lysotracker accumulation (Supplementary Fig. 3a) and increased DQ BSA degradation (Supplementary Fig. 3b). DexoMAG did not perturb mTORC1 signalling (Supplementary Fig. 3c) or cell proliferation (Supplementary Fig. 3d). Starting with live cells, magnetic isolation of DexoMAG-loaded lysosomes took about 30 min and yielded an organelle fraction that was highly enriched in lysosomal luminal and membrane proteins (Fig. 3a) and intact, as assessed by liberation of cathepsin activity upon detergent treatment (Supplementary Fig. 3e). Raptor, an mTORC1-specific subunit, and RagA, which activates mTORC1 at lysosomal membranes, were also highly enriched (Fig. 3a). By contrast, the mTORC2-specific subunit Rictor and markers for other organelles were depleted.

Proteins were quantitated in lysosomal fractions and cell lysates (post-nuclear supernatants) using liquid chromatography–mass spectrometry and label-free quantification. On average, lysosomal proteins were 45-fold enriched in lysosomal fractions over non-lysosomal proteins; V-ATPase and mTORC1 subunits were even enriched by 62-fold and 80-fold, respectively, validating our organelle enrichment approach (Fig. 3b, Supplementary Data 1). To determine mTORC1-regulated changes in lysosomal proteins, lysosomes were enriched from cells in which mTORC1 was inactivated by amino acid starvation or acutely activated by amino acid restimulation. Intriguingly, V-ATPase subunits that are part of the $V_1$ domain subcomplex were among the most strongly decreased proteins in lysosomal fractions of amino acid-restimulated cells (Fig. 3c). To confirm these results, we prepared organelle fractions through centrifugation of dextran-weighted lysosomes and examined the abundance of V-ATPase subunits by western blot. When amino acid-starved cells were subjected to amino acid restimulation, the abundance of membrane-associated V-ATPase $V_1$ domain subunits V1A and V1E1 decreased (Fig. 3d, e). This is consistent with previous findings that the peripheral $V_1$ domain reversibly associates with the membrane-integral $V_o$ domain to assemble a functional proton pump[4,5,23]. Conversely, V1A and V1E1 levels increased in membrane fractions upon torin 1 treatment (Fig. 3d, e). The $V_o$ domain subunit Vod1 did not display significant changes in response to either amino acid restimulation or torin 1. Together, these results suggested that reversible assembly of $V_1$ domains with membrane-integral $V_o$ domains is regulated by mTORC1.

To monitor V-ATPase assembly at lysosomes over time, we established a live cell imaging assay using $V_o$ and $V_1$ domain subcomplexes labelled with different fluorescent proteins (Fig. 4a). The $V_1$ domain was labelled by expression of V1B2-mNeonGreen. In resting cells, most V1B2-mNeonGreen displayed a diffuse cytosolic localization. However, torin 1 treatment caused a dramatic accumulation of V1B2 in punctate structures that appeared to be lysosomes (Fig. 4b, Supplementary Movie 1). To directly follow the recruitment of $V_1$ domains to lysosomes, we labelled the lysosomal pool of $V_o$ domains by expression of Voa3-mScarlet, which colocalized with the ubiquitous lysosomal membrane protein Lamp1 (Lamp1-mNeonGreen) regardless of the status of mTORC1 activity (Supplementary Fig. 4a–c). Torin 1 treatment caused rapid accumulation of V1B2 specifically at Voa3-containing organelles (Fig. 4c, Supplementary Fig. 4d, e). Within 60 min of mTORC1 inhibition, V1B2 levels had increased throughout the lysosomal population (Fig. 4d, e). Conversely, when amino acid-starved cells were restimulated with amino acids, most V1B2

dissociated from lysosomal $V_o$ within 20 min (Fig. 4f, Supplementary Movie 2). Rapamycin did not induce lysosomal movement of V1B2, consistent with the rapamycin insensitivity of lysosomal acidification (Supplementary Fig. 5a). Overall, these results show that the V-ATPase $V_1$ domain rapidly and reversibly moves to and from $V_o$ domain-containing lysosomes in response to changes in mTORC1 activity.

Recruitment of $V_1$ domains to $V_o$ domains at lysosomal membranes leads to assembly of active V-ATPases, which acidify the lysosomal lumen[4,5]. Thus, we examined the interrelation of lysosomal $V_1$ domain recruitment and lysosomal acidification in response to mTORC1 inhibition. Torin 1 caused the movement of V1B2 to lysosomes that accumulated cresyl violet over a comparable timespan (Fig. 4g). An increase in lysosomal $V_1$ domains and cresyl violet was also observed with a different fluorescent fusion protein, V1A-eGFP (Supplementary Fig. 5b), and in human carcinoma cells (Supplementary Fig. 5c). Finally, we examined the role of assembly factors that have been implicated in the regulation of the mammalian V-ATPase. The Rab7 effector, RILP, has been reported to promote endosomal recruitment and stability of the V-ATPase V1G1 subunit[24]. However, genetic ablation of RILP did not decrease the torin 1-induced increase in lysosomal V1B2 and cresyl violet (Supplementary Fig. 5d). This suggests that RILP is dispensable for V-ATPase assembly in response to mTORC1 inactivation. Next, we examined DMXL1/2, which appear to be functionally related to the yeast V-ATPase assembly factor, the RAVE complex[25]. Genetic ablation of DMXL1/2 strongly suppressed lysosomal V1B2 and cresyl violet levels under basal and mTORC1-inhibited conditions (Supplementary Fig. 5d). Overall, these results show that mTORC1 regulates lysosomal acidification through reversible, DMXL1/2-dependent recruitment of the V-ATPase $V_1$ domain to and from lysosomal membranes.

## The V-ATPase V1 domain reversibly associates with the cytosolic chaperonin TRiC

To understand why a large fraction of the V-ATPase $V_1$ domain was present in the cytosol of cells with high mTORC1 activity, we determined $V_o$ and $V_1$ domain-specific interaction partners using co-immunoprecipitation (Co-IP). Flag-tagged Voa3 or HA-tagged V1B2 were immunoprecipitated from cell lysates, which efficiently coprecipitated other subunits of the $V_o$ and $V_1$ domain subcomplexes (Supplementary Fig. 6a). V-ATPase-interacting proteins were then quantified using liquid chromatography–mass spectrometry and SILAC (Supplementary Fig. 6b, Supplementary Data 2). Almost all $V_o$ and $V_1$ domain subunits were robustly quantified in Voa3 Co-IPs (Fig. 5a). Voa3 Co-IPs also contained other known V-ATPase interaction partners: Tmem199, Ccdc115 and Atp6ap1, which are required for $V_o$ domain assembly in the endoplasmic reticulum[26,27], subunits of the ragulator complex (Lamtor 1 and 3), which interacts with the V-ATPase at lysosomal membranes[28], and Tmem55b, which interacts with the V-ATPase and ragulator[29]. In addition, we identified a previously not recognized interaction with flotillin-1 and flotillin-2, two membrane-scaffolding proteins that were also highly enriched in our lysosomal fractions ($\log_2$ fold change >6). Tmem199, Ccdc115 and Atp6ap1 were not detected in V1B2 Co-IPs, consistent with their function in assembly of newly synthesised $V_o$ domains[26,27]. By contrast, the lysosomal V-ATPase interaction partners, ragulator, Tmem55b and flotillins, were robustly identified in V1B2 Co-IPs (Fig. 5b). Moreover, all subunits of the cytosolic chaperonin tailless complex polypeptide 1 ring complex (TRiC), Cct1–Cct8, were coimmunoprecipitated with V1B2 (Fig. 5b)[30]. Strikingly, TRiC subunits were not identified in Voa3 Co-IPs (Supplementary Fig. 6b). This suggests that the $V_1$ domain can associate either with the membrane-integral $V_o$ domain or with cytosolic TRiC.

We next examined whether the interaction partners of the $V_1$ and $V_o$ domains changed in response to mTORC1 signalling. Voa3 and V1B2 were immunoprecipitated from amino acid-starved and amino acid-restimulated cells, and co-precipitating proteins

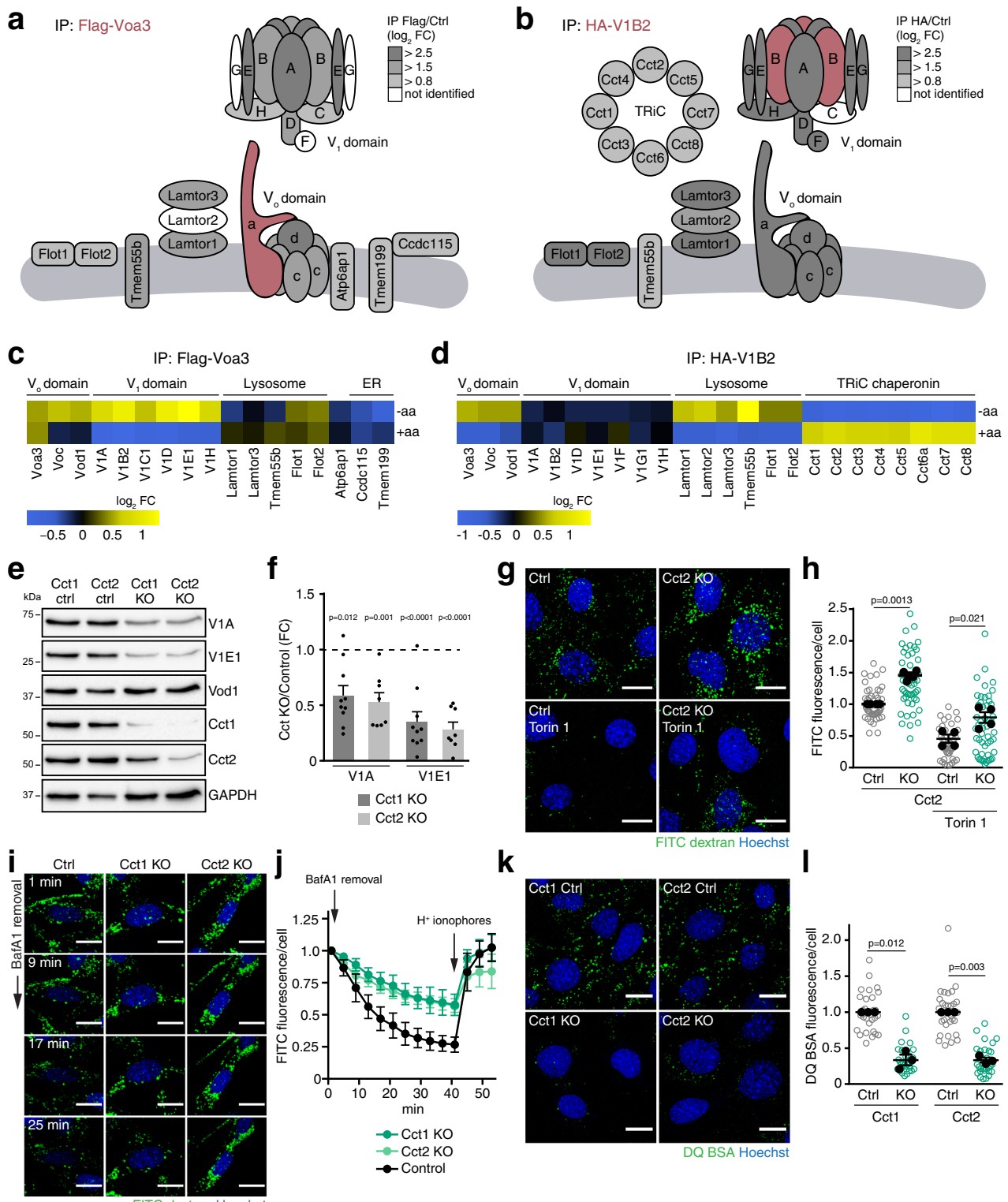

quantified using liquid chromatography–mass spectrometry and SILAC. Voa3 Co-IPs contained comparable levels of other $V_o$ domain subunits, ragulator components and ER-resident assembly factors in the different nutrient conditions (Fig. 5c; Supplementary Data 2). By contrast, Voa3 coimmunoprecipitated $V_1$ domain subunits at increased levels upon amino acid starvation, and conversely at decreased levels upon amino acid restimulation, confirming that amino acid levels regulate reversible V-ATPase assembly. Intriguingly, the $V_1$ domain displayed a reciprocal change in its lysosomal and cytosolic interaction partners. Upon amino acid starvation,

V1B2 coimmunoprecipitated increased levels of $V_o$ domain subunits and other lysosomal proteins, but decreased levels of TRiC (Fig. 5d). Conversely, upon amino acid stimulation V1B2 coimmunoprecipitated increased levels of TRiC, but decreased levels of $V_o$ and other lysosomal proteins. Consistently, Flag-tagged variants of the TRiC subunits Cct1 and Cct2 coimmunoprecipitated higher amounts of the $V_1$ domain subunits V1A and V1B2 when cells were restimulated with amino acids (Supplementary Fig. 6c). Overall, these findings show that the V-ATPase $V_1$ domain reversibly associates with different interaction partners: Upon amino acid starvation, the $V_1$

**Fig. 5 | The V-ATPase V₁ domain reversibly associates with the cytosolic chaperonin TRiC. a, b** Cartoon summarizing SILAC quantification data of proteins in Co-IPs from MEFs with **a** Flag-Voa3, **b** HA-V1B2. **c, d** Changes in protein abundance in SILAC Co-IPs with **c** Flag-Voa3, **d** HA-V1B2 for 1 h aa starvation (−aa) versus full medium, or for 1 h aa starvation + 30 min aa restimulation versus 1 h aa starvation (+aa). **e** Changes in V-ATPase subunit abundance in Cct1 and Cct2-deficient MEFs, analysed by western blot. **f** Western blot quantification of V-ATPase subunit abundance changes in Cct1 and Cct2-deficient cells as in **e**. Data are normalized replicate mean ± SEM. The dashed line indicates protein levels in control cells. **g** Fluorescence quenching of lysosomally loaded FITC-dextran in Cct2-deficient MEFs ± 1 h torin 1 [400 nM]. **h** Quantification of FITC fluorescence quenching of Cct2-deficient cells ± torin 1 as in **g**. Data are normalized replicate mean ± SEM (closed circles) and fields of view (open circles; ≥10 per replicate). **i** Fluorescence quenching of lysosomally loaded FITC-dextran in Cct1 and Cct2-deficient MEFs + 1 h torin 1 [400 nM] after removal of the V-ATPase inhibitor bafilomycin A1 [20 nM]. Scale bars = 20 μm. **j** Quantification of FITC fluorescence quenching over time in cells treated as in **i**. Data are normalized replicate mean ± SEM (12 fields of view per replicate). **k** DQ BSA degradation in Cct1 and Cct2-deficient MEFs after 6 h DQ BSA uptake + torin 1 [400 nM]. Scale bars = 20 μm. **l** Quantification of DQ BSA fluorescence of Cct1 and Cct2-deficient cells treated as in **k**. Data are normalized replicate mean ± SEM (closed circles) and fields of view (open circles; ≥10 per replicate). **a–d** n = 4, **f** n = 8 (Cct1 KO) or n = 10 (Cct2 KO), **h** n = 4, **j** n = 3, **l** n = 3 biologically independent experiments. In **f**, p-values were calculated using a one sample t-test with a hypothetical mean of 1. In **h**, **l**, p-values were calculated using a two-tailed unpaired t-test with Welch correction. Source data are provided as a Source data file.

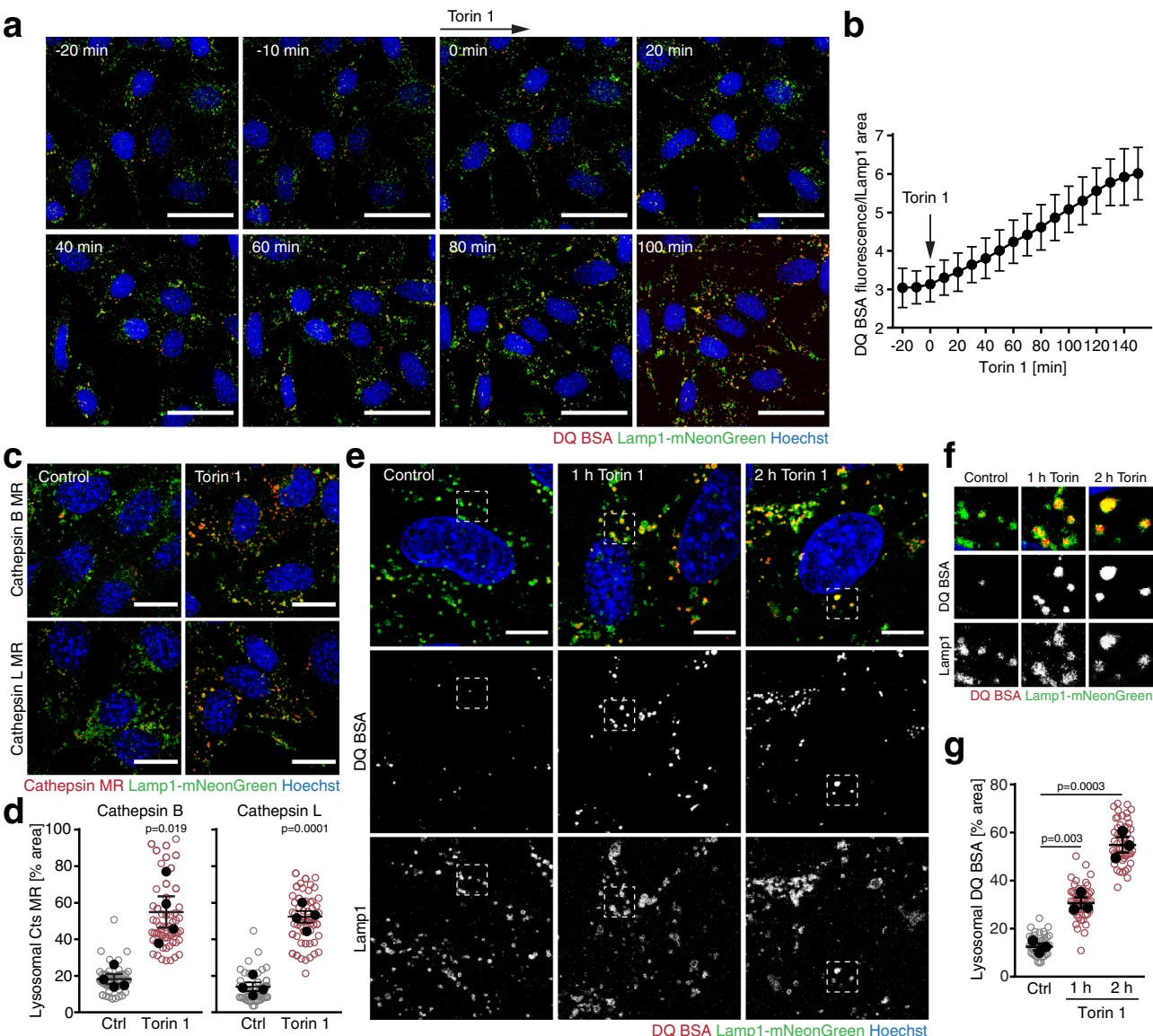

**Fig. 6 | Rapid mobilization of lysosomal proteolytic capacity in response to mTORC1 inactivation. a** Degradation of lysosomally preloaded DQ BSA in MEFs upon torin 1 treatment [400 nM]. Scale bars = 20 μm. **b** Quantification of DQ BSA in Lamp1-containing lysosomes of cells shown in **a**. Data are mean ± SD (25 fields of view). **c** Lysosomal degradation of magic red substrates of cathepsins B and L in MEFs after 1 h pretreatment ± torin 1 [400 nM]. Scale bars = 20 μm. **d** Fraction of lysosomes with high cathepsin B/L activity in cells treated as in **c**. Data are replicate mean ± SEM (closed circles) and fields of view (open circles; ≥10 per replicate).

**e** Degradation of lysosomally preloaded DQ BSA in MEFs after 0–2 h torin 1 [400 nM]. Scale bars = 10 μm. **f** Magnification of areas highlighted in **e**. **g** Fraction of lysosomes with high DQ BSA degradation in cells treated as in **e**. Data are replicate mean ± SEM (closed circles) and fields of view (open circles; ≥10 per replicate). **b** One representative of n = 3 biologically independent experiments; **d** n = 4, **g** n = 3 biologically independent experiments. p-values were calculated using a two-tailed unpaired t-test with Welch correction. Source data are provided as a Source data file.

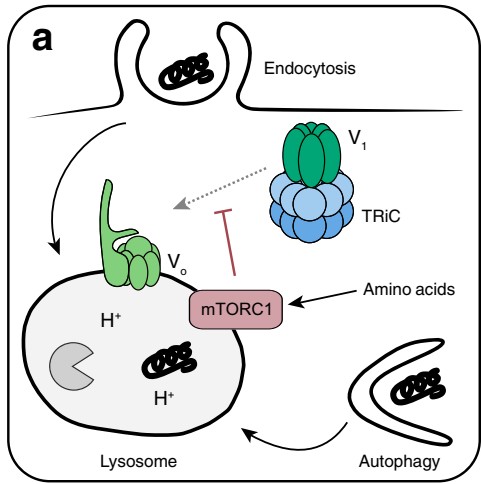
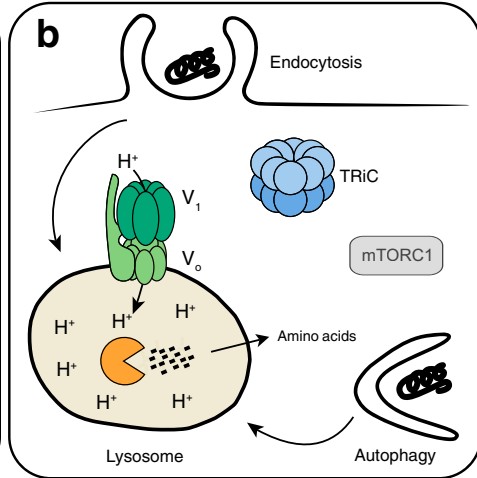

**Fig. 7 | mTORC1 controls lysosomal catabolic activity by regulating V-ATPase assembly.** Model for the regulation of lysosomal catabolic activity by mTORC1. **a** Under amino acid-replete conditions, a large fraction of V-ATPase $V_1$ domains is stabilized in the cytosol by association with the chaperonin TRiC. Consequently, lysosomes display an elevated pH and low catabolic activity. **b** Upon amino acid starvation and the ensuing decline in mTORC1 signalling, $V_1$ domains move to lysosomes and assemble with $V_o$ domains into active proton pumps. The resulting acidification of the lysosomal lumen increases proteolytic activity and initiates degradation of macromolecular contents throughout the organelle population.

domain assembles with the $V_o$ domain at lysosomal membranes, and reassociates with TRiC in response to amino acid stimulation.

### TRiC is required for V1 domain stabilization and lysosomal acidification

To address the relevance of the V-ATPase $V_1$ domain interaction with TRiC, we genetically ablated the TRiC subunits Cct1 and Cct2 using CRISPR/Cas9. Loss of Cct1 or Cct2 caused a significant decrease in the $V_1$ domain subunits V1A and V1E1 (Fig. 5e, f). Consistently, Cct1 and Cct2-deficient cells displayed a strong reduction in the levels of V1B2-mNeonGreen (Supplementary Fig. 7a, b). However, Cct1 and Cct2-deficient cells did not exhibit a decrease in the levels of Vod1 (Fig. 5e), or apparent changes in the abundance and morphology of Lamp1-mNeonGreen-containing lysosomes (Supplementary Fig. 7c, d). These results suggest that the V-ATPase $V_1$ domain is stabilized by association with TRiC.

Next, we examined the consequence of TRiC depletion for lysosomal acidification and proteolytic activity. To monitor lysosomal acidification, we loaded lysosomes with dextran-conjugated FITC, whose fluorescence becomes progressively quenched at acidic pH. Under basal conditions, FITC fluorescence was less quenched in Cct1 and Cct2-deficient cells than in control cells (Fig. 5g, h). In response to torin 1 treatment, FITC fluorescence remained elevated in Cct1 and Cct2-deficient cells, but was strongly quenched in control cells. Thus, loss of Cct1 or Cct2 impairs lysosomal acidification under basal and mTORC1-inhibited conditions. To examine V-ATPase-mediated acidification of lysosomes directly, we monitored the re-acidification rate of lysosomes whose luminal pH was neutralized. Cells with FITC-dextran-loaded lysosomes were treated with the V-ATPase inhibitor bafilomycin A1 to raise lysosomal pH and with torin 1 to increase V-ATPase assembly. When bafilomycin A1 was removed, the resumption of V-ATPase activity rapidly led to FITC quenching (Fig. 5i, j). In striking contrast, FITC was quenched slowly and to a lesser extent in Cct1 and Cct2-deficient cells, indicative of a decrease in V-ATPase-mediated proton pumping. Dissipation of lysosomal pH de-quenched fluorescence to the same level as in control cells, indicating that lysosomes contained FITC-dextran in comparable quantities. Finally, we examined the activation of DQ BSA degradation in response to mTORC1 inhibition. Torin 1-induced DQ BSA degradation was strongly blunted in Cct1 and Cct2-deficient cells (Fig. 5k, l). DQ BSA degradation was restored by ectopic expression of Cct1 or Cct2 rescue constructs (Supplementary Fig. 7e). Overall, these results suggest that TRiC stabilizes the cytosolic pool of V-ATPase $V_1$ domains, which is recruited to lysosomes in response to mTORC1 inactivation to initiate lysosomal acidification and protein degradation.

The TRiC subunit Cct2 is phosphorylated on S260 by the mTORC1 effector S6 kinase (S6K)[31]. This raised the question whether mTORC1 regulated V-ATPase assembly through S6K-mediated phosphorylation of Cct2. To address this, we first asked whether S6K inhibition recapitulated the effects of mTORC1 inhibition. Unlike torin 1, the S6K inhibitor LY2584702 did not increase lysosomal cresyl violet signal or DQ BSA degradation (Supplementary Fig. 7f–i). Next, we examined the relevance of Cct2 S260 phosphorylation. To this end, phosphorylation-deficient Cct2 S260A and phosphomimetic Cct2 S260D variants were ectopically expressed in cells where endogenous Cct2 was depleted using CRISPR/Cas9. Loss of Cct2 caused a strong defect in cell proliferation, which was fully rescued by Cct2 wild type and S260D, but only slightly alleviated by Cct2 S260A (Supplementary Fig. 7j). However, Cct2 wild type, S260A and S260D restored DQ BSA degradation to a similar extent, both at basal levels and at increased levels in response to torin 1 (Supplementary Fig. 7k). Consistently, all Cct2 variants rescued the abundance of V-ATPase $V_1$ domain subunits (Supplementary Fig. 7l). Thus, mTORC1 suppresses lysosomal acidification and proteolysis through a mechanism that does not involve S6K-mediated regulation of TRiC.

### mTORC1 inhibition induces rapid activation of lysosomal catabolism

The above data indicate that mTORC1 exerts direct control over lysosomal catabolism by suppressing V-ATPase assembly. To further investigate this, we determined the impact of mTORC1 inactivation on catabolic activity throughout the cellular population of lysosomes. Lysosomes were labelled in MEFs by endogenous tagging of Lamp1 with mNeonGreen. To examine the rate at which lysosomal catabolism was activated in response to mTORC1 inhibition, we preloaded lysosomes with DQ BSA and followed DQ BSA fluorescence dequenching over time. While DQ BSA fluorescence did not change in resting cells, DQ BSA fluorescence began to increase in response to torin 1 within 30 min, indicating activation of lysosomal proteolysis (Fig. 6a, b). We noted that most lysosomes in resting cells displayed little DQ BSA signal, whereas a subset of lysosomes displayed substantial DQ BSA

fluorescence. Thus, proteolytic activity of lysosomes was heterogeneous across the organelle population, consistent with previous observations[32]. This raised the question whether mTORC1 inhibition increased proteolysis in a subset of lysosomes, or activated lysosomal catabolism across the lysosomal population. To distinguish these possibilities, we quantified the fraction of Lamp1-positive organelles that displayed high protease activity upon mTORC1 inhibition using cathepsin MR substrates. In control cells, less than 20% of the lysosomal population displayed discernible cathepsin activity (Fig. 6c, d). In stark contrast, after 1 h of torin 1 treatment about 60% of lysosomes exhibited strong MR fluorescence. Consistently, only a minor fraction of lysosomes degraded DQ BSA in control cells, but torin 1 caused a strong increase in DQ BSA fluorescence across a large fraction of lysosomes (Fig. 6e–g). Thus, most lysosomes degrade little protein when mTORC1 is active. When mTORC1 activity declines, cells respond with a rapid increase in proteolytic activity throughout the lysosomal population.

## Discussion

The present work demonstrates that mTORC1 prevents the nutritional use of extracellular proteins by suppressing the catabolic activity of lysosomes (Fig. 7). Under nutrient-rich conditions where mTORC1 is active, many lysosomes display an elevated pH and low protease activity. Concomitantly, a large fraction of the peripheral V-ATPase $V_1$ domain resides in the cytosol in an assembly-competent pool that is stabilized through association with the chaperonin TRiC. When mTORC1 activity declines, $V_1$ domains move to the lysosome and assemble with $V_o$ domains into functional proton pumps to acidify the organelle. The resulting activation of proteases rapidly initiates the degradation of protein contents that were accumulated in the lysosomal lumen. These findings establish direct control of lysosomal catabolic activity by mTORC1 as a principle of metabolic regulation in mammalian cells.

Lysosomes are the point of convergence for the cellular machinery that integrates nutrient availability and growth signals to regulate mTORC1[3]. Amino acid levels are communicated to mTORC1 through the ragulator complex, which controls lysosomal recruitment and subsequent activation of mTORC1 through the Rag GTPases[10–12]. Ragulator also interacts with the V-ATPase in an amino acid-sensitive manner[28]. Activation of mTORC1 by amino acids depends on the V-ATPase, but this process does not appear to involve lysosomal acidification. The above findings indicate that the V-ATPase $V_o$ domain interacts with ragulator constitutively, whereas the $V_1$ domain reversibly associates with $V_o$-ragulator complexes in an mTORC1-dependent process. Thus, the V-ATPase is part of the upstream machinery that regulates mTORC1, as well as a downstream effector through which mTORC1 controls lysosomal function. Previously known mechanisms through which mTORC1 regulates lysosomal catabolism act indirectly −cytosolic initiation of autophagosome formation and nuclear expression of lysosomal genes[10,11]. By contrast, regulation of V-ATPase assembly is a mechanism through which mTORC1 exerts direct control over acidification and catabolic activity of lysosomes. The activation of mTORC1 at lysosomal membranes may thus serve to closely coordinate lysosomal function with nutrient availability.

Reversible association and dissociation of $V_1$ and $V_o$ domain subcomplexes is an evolutionarily conserved mechanism that regulates V-ATPase proton pumping activity[4,5]. At the molecular level, V-ATPase assembly has been studied primarily in yeast, where the RAVE complex binds cytosolic $V_1$ domains and catalyses their assembly with membrane-bound $V_o$ domains. In mammalian cells, several proteins have been implicated in V-ATPase assembly, including DMXL1/2, which appear to be functional equivalents of yeast RAVE[25]. V-ATPase assembly and disassembly is controlled by metabolic signals. Yeast $V_1$ and $V_o$ domains rapidly dissociate in response to glucose starvation in a protein kinase A-regulated process[33]. Conceivably, decreasing the

number of active proton pumps allows starving yeast to reduce ATP consumption. The mammalian V-ATPase is also sensitive to changes in nutrient availability, but its assembly increases in response to starvation[23,34]. Although many nutrient signals are transmitted by mTORC1, its role in the regulation of V-ATPase assembly has been unclear. Acute rapamycin treatment does not recapitulate starvation responses (ref. 23, the present study), and long-term rapamycin treatment decreases V-ATPase assembly during dendritic cell maturation[35]. These apparent discrepancies were resolved by the use of mTOR kinase inhibitors, which potently induce lysosomal V-ATPase assembly and acidification. Thus, the present study identifies mTORC1 as the signalling pathway that regulates DMXL1/2-dependent V-ATPase assembly at lysosomes in response to amino acid availability.

Our work further reveals an amino acid-regulated interaction between the V-ATPase $V_1$ domain and the cytosolic chaperonin TRiC. Intriguingly, $V_1$–TRiC and $V_1$–$V_o$ interactions display reciprocal changes, consistent with the shuttling of $V_1$ domains between cytosol and lysosomes. Loss of TRiC destabilizes $V_1$ domains, suggesting that TRiC maintains a large cytosolic pool of $V_1$ domains, which can rapidly assemble proton pumps at lysosomes in response to metabolic cues. How mTORC1 regulates the balance between the $V_1$–TRiC and $V_1$–$V_o$ interactions at the molecular level remains an outstanding question. The TRiC subunit Cct2 is phosphorylated on S260 by the mTORC1-activated kinase S6K[31], and TRiC was recently identified as an mTORC1 effector that stabilizes the $m^6A$ RNA methyltransferase complex[36]. However, our results argue against a direct regulation of the $V_1$–TRiC interaction by the mTORC1 pathway. Neither S6K signalling nor Cct2 S260 phosphorylation are involved in the process through which mTORC1 suppresses lysosomal acidification and protein degradation. This suggests that mTORC1 promotes $V_1$ domain association with TRiC indirectly, perhaps by blocking $V_1$ domain recruitment into V-ATPase complexes.

Previous work showed that inhibition of mTORC1 or its upstream activator, the serine/threonine kinase Akt, leads to increased lysosomal amino acid generation from extracellular proteins[8,37]. As a corollary, mTOR inhibitors promote the growth of cells that rely on extracellular proteins as an obligatory nutrient source, such as cancer cells that reside in nutrient-poor tumour microenvironments. V-ATPase inhibition is likely an efficient strategy to block the growth-promoting action of mTOR inhibitors in such metabolic contexts. However, attempts to use V-ATPase inhibitors in cancer therapy were discouraged by their high toxicity due to the pleiotropic functions of V-ATPases in the endomembrane system[38]. Our work suggests targeting the lysosomal V-ATPase assembly pathway as an alternative strategy to specifically suppress lysosomal catabolism without broad perturbation of cellular functions.

Mammalian cells adapt to nutrient shortages by supplying increased quantities of macromolecular substrates to lysosomal catabolism as alternative nutrient sources. In response to starvation, cells induce the formation of autophagosomes to sequester intracellular constituents[9], and transcriptionally upregulate macropinocytosis to promote uptake of extracellular macromolecules[39,40]. Lysosomal flux of macromolecules from the extracellular environment can also be increased by growth factor signalling. For example, the formation of macropinosomes is orchestrated by Ras and PI3-kinase, and the expression of various endocytic receptors is enhanced by their downstream effectors Akt and mTORC1[41,42]. However, the present work suggests that macromolecular contents which accumulate in the lysosomal lumen are only inefficiently degraded in nutrient-replete conditions. Rather, the activation of lysosomal catabolism through V-ATPase assembly and luminal acidification constitutes a distinct process that is regulated by cellular starvation responses. By controlling lysosomal acidification, cells are able to adjust lysosomal catabolic activity within less than one hour. The resulting increase in endocytic and autophagic cargo degradation generates sufficient amino acids to

reactivate mTORC1, which may constitute a feedback mechanism to limit lysosomal catabolism[8,43]. Starving cells also enhance their degradative capacity over time by increasing expression of lysosomal genes including V-ATPase subunits[14,15,44]. Conceivably, acute activation of lysosomal proteases and long-term increase in lysosomal abundance allows dynamic adjustment of lysosomal catabolism over different time scales. It was noted previously that a substantial fraction of lysosomes in mammalian cells, although containing degradative enzymes, has an elevated pH and is catabolically inactive[32]. Our work shows that such lysosomes become proteolytically active upon termination of mTORC1 signalling. The mobilization of latent lysosomal catabolic capacity may enable cells to rapidly adjust metabolic activities to fluctuations in nutrient supply.

## Methods

### Reagents
Antibodies were from Cell Signaling (9452 4E-BP1, 39749 SQSTM1 (D1Q5S), 3724 HA-tag (C29F4), 2920 AKT (pan) (40D4), 4060 AKT pS473 (D9E), 12238 Calreticulin (D3E6), 13192 Golgin-97 (D8P2K), 4357 RagA (D8B5), 2280 Raptor (24C12), 9476 Rictor (D16H9), 2708 S6K (49D7), 9234 S6K pT389 (108D2), 8054 ULK1 (D8H5), 6888 ULK1 pS757, 2217 S6 (5G10), 2215 S6 pS240/244, 4661 VDAC (D73D12), 12994 ATG5 (D5F5U), 2276 myc-tag (9B11)), R&D Systems (AF965 cathepsin B, AF1515 cathepsin L, AF1029 cathepsin D, AF1033 cathepsin X/Z/P), Invitrogen (MA3-026 CCT1 (91A)), Sigma-Aldrich (A5441 β-actin, PA22129 Pex19), GenScript (A00187-200 Flag-tag), Proteintech (15280 ATP6V1E1), Origene (TA802519 GAPDH (OTI2D9)) and Abcam (ab25245 Lamp1 (1D4B), ab202899 ATP6V0d1 (EPR18320-38), ab199326 ATP6V1A (EPR19270), ab92746 CCT2 (EPR4084)). The LC3 antibody was a kind gift from Tullia Lindsten. Secondary antibodies were from Life Technologies (31402 HRP-linked anti-goat), Cytiva (NA931 Amersham ECL Mouse IgG, NA934 Amersham ECL Rabbit IgG), and Sigma-Aldrich (A10549 HRP-linked anti-rat). Primary antibodies were used at 1:1000 dilution, secondary antibodies at 1:5000 dilution.

Oregon Green 488 dextran (70 kDa), Alexa Fluor 568 dextran (10 kDa), FITC dextran (10 kDa), Lysosensor Yellow/Blue DND-160 dextran (10 kDa), DQ Green BSA, DQ Red BSA, Hoechst 33342, and Lysotracker Red were from Life Technologies, Magic Red Cathepsin B (KF17308) and Magic Red Cathepsin L (KF17312) from Immunochemistry Technologies, and cresyl violet acetate from MP biomedicals (150727).

Inhibitors were from Tocris (nigericin, torin 1), Cayman Chemical (AZD8055, bafilomycin A1), EMD Chemicals (rapamycin), Santa Cruz Biotechnology (cycloheximide), and Selleckchem (Monensin, LY2584702). Aprotinin (Thermo Fisher Scientific), benzaminidine (Sigma-Aldrich), E64, AEBSF, leupeptin and pepstatin (Serva) or Halt Protease and Phosphatase Inhibitor Mix (Thermo Fisher Scientific) were used as protease inhibitor cocktails. Sodium orthovanadate, sodium fluoride, sodium pyrophosphate and sodium glycerophosphate (Sigma-Aldrich) were used as phosphatase inhibitors.

### Cell culture
Human MIA PaCa-2 (ATCC CRL-1420), A549 (ATCC CCL-185), MRC-5 (ATCC CCL-171) and SV40 large T antigen-immortalized mouse embryonic fibroblasts (MEFs)[8] were cultured in DMEM/F-12 supplemented with 10% FBS and 2 mM glutamine. HEK 293T (ATCC CRL-3216) were cultured in DMEM supplemented with 10% FBS and 2 mM glutamine. All cell lines were maintained at 37 °C and 5% CO$_2$, routinely tested for mycoplasma contamination (MycoAlertTM Mycoplasma Detection, Lonza) and authenticated by Single Nucleotide Polymorphism (SNP) typing by Multiplexion or by sequencing analysis. Cell culture reagents were from Gibco.

Experiments with amino acid-deficient cell culture media were performed with amino acid-free, glucose-free DMEM/F-12 (USBiological D9807-11) and dialysed FBS (Gibco). Amino acids (where indicated),

glucose and bicarbonate (Sigma-Aldrich) were re-added to the concentrations of standard DMEM/F-12; the pH was adjusted to 7.3 with HCl.

### Lentivirus production and transduction
HEK 293T cells were co-transfected with lentiviral vector, pCMVR8.74 (Addgene 22036) and pCMV-VSV-G (Addgene 8454) using polyethylenimine (PEI, MW 25000, Polysciences). The virus-containing supernatants were cleared by filtration through a 0.45 μm PES filter. Target cells were transduced by addition of viral supernatant and 10 μg/ml polybrene (Sigma-Aldrich).

### Generation of gene knockouts
For the generation of clonal inducible Cas9 (iCas9) MEFs, cells were sequentially transduced with pRRL-SFFV-rtTA3-IRES-EcoR-PGK-HygroR and pLentiv2-TRE3G-Cas9-P2A-BFP[45], selected with hygromycin B and FACS-sorted into 96-well plates using a FACSAria III cell sorter to obtain single-cell derived clones. For generation of inducible knockout (iKO) cells, iCas9 MEFs were transduced with Dual-hU6-sgRNA-mU6-sgRNA-Ef1a-Thy1.1-P2A-NeoR or Dual-hU6-sgRNA-mU6-sgRNA-EF1αs-BFP harbouring sgRNAs targeting a single gene[45]. sgRNA-positive cells were selected with neomycin or FACS sorting of BFP$^+$ cells. Cas9 expression was induced with 0.4 μg/ml doxycycline for 3 days. Controls were Cas9 and sgRNA-expressing cells without doxycycline treatment. For constitutive gene knockouts, MEFs were transduced with LentiCRISPR v2 (Addgene 52961) harbouring sgRNAs against target genes or control sgRNAs. sgRNA sequences are listed in Supplementary Table 1.

### Generation of V-ATPase and TRiC expression constructs
Open reading frames of human *ATP6V1B2* and *ATP6Voa3* (*TCIRG1*) were obtained from the Genomics and Proteomics Core Facility at the DKFZ. cDNAs were amplified using primers that introduced an N-terminal 3xHA-tag or 3xFlag-tag. To generate fluorescent protein fusions, mNeonGreen and mScarlet-I were amplified from pMaCTag-Z07 (Addgene 124790) and pMaCTag-Z11 (Addgene 120054), respectively. HA-V1B2 and HA-V1B2-mNeonGreen were cloned into pLV-EF1a-IRES-BlastR (Addgene 85133), Flag-Voa3 and Flag-Voa3-mScarlet-I were cloned into pLV-EF1a-IRES-HygroR (Addgene 85134). For Cct1 and Cct2 rescue experiments, cDNAs for murine *Cct1* (*Tcp1*) and *Cct2* were synthesized by Genewiz and subsequently cloned with an N-terminal Myc-tag into pLV-EF1a-IRES-HygroR (Addgene 85134). At least 4 synonymous nucleotide changes were introduced in the sgRNAs biding sites to make the expression constructs resistant to Cas9 editing. Cct2 was additionally mutated to Cct2 S260A and Cct2 S260D by site-directed mutagenesis (Q5 Site-Directed Mutagenesis Kit, NEB). Primers for cDNA cloning were designed using the NEBbuilder Assembly Tool (nebuilder.neb.com). Cloning was performed using the HiFi DNA Assembly Master Mix (NEB).

### Generation of endogenously tagged Lamp1-mNeonGreen cell lines
The generation of endogenously tagged Lamp1 knock-in cells was performed as previously described[46]. A mNeonGreen construct for C-terminal tagging of Lamp1 was amplified by PCR from pMaCTag-Z07 (Addgene 124790) with primers designed using the online oligo design tool (www.pcr-tagging.com) and modified with phosphorothioate groups at the five terminal 5′ end nucleotides (Supplementary Table 2). PCR amplification was performed using Velocity HiFi-polymerase (Bioline BIO-21098) and self-made PCR buffer (200 mM Tris-HCl pH 8.8, 100 mM (NH$_4$)$_2$SO$_4$, 500 mM KCl, 1% Triton X-100, 1 mg/ml BSA, 20 mM MgCl$_2$, 500 mM betaine). Template plasmid was removed by digestion with DnpI (NEB R0176S). Plasmids for PCR tagging were a kind gift from Michael Knop. MEFs were co-transfected with 500 ng of the PCR cassette and 500 ng pcDNA3.1-hAsCpf1 (Addgene 89353)

using Lipofectamine 3000 (ThermoFisher Scientific). After three passages to deplete non-integrated tagging constructs, mNeonGreen-positive cells were sorted into 96-well plates using a FACSAria III cell sorter to obtain single-cell derived clones. Successful tagging of Lamp1 with mNeonGreen was confirmed by western blotting and confocal microscopy.

## Western blotting

Cells were rinsed with ice-cold PBS, lysed in ice-cold lysis buffer (50 mM HEPES pH 7.4, 40 mM $NaCl_2$, 2 mM EDTA, 1 mM sodium orthovanadate, 50 mM sodium fluoride, 10 mM sodium pyrophosphate, 10 mM sodium glycerophosphate, 1% Triton X-100, 1x Halt protease and phosphatase inhibitor cocktails) for 15 min and soluble lysate fractions isolated by centrifugation at $18,000 \times g$ for 5 min. Protein concentrations were determined with the Pierce BCA Protein Assay (Thermo Fisher Scientific) and equal protein amounts analysed by SDS gel electrophoresis and western blotting following standard protocols. LC3 was blotted on PVDF membranes, all other proteins on nitrocellulose membranes. When experiments were analysed using multiple antibodies on different membranes, sample preparation, electrophoresis and Western blotting were performed in parallel under identical conditions. Blots were imaged using a ChemiDoc Touch imaging system (BioRad) operated by Image Lab (v3.0.1.14).

Rapid preparation of lysosome-containing membrane fractions was performed using dextran-loading of lysosomes and centrifugation. MEFs were incubated overnight with 2 mg/ml 10 kDa Dextran (Sigma-Aldrich). After two media washes, dextran was accumulated in lysosomes by a 4 h chase in fresh media. Next, cells were rinsed with ice-cold PBS, resuspended in homogenization buffer (20 mM HEPES (pH 7.5), 125 mM KCl, 50 mM sucrose, 1 mM EDTA, protease and phosphatase inhibitors), and homogenized by 12 strokes with a KIMBLE Dounce tissue grinder with a large clearance pestle (Sigma-Aldrich). Then, nuclei were removed by centrifugation at $800 \times g$ for 5 min and membranes pelleted from post-nuclear supernatants by centrifugation at $18,000 \times g$ for 20 min. Resulting pellets were resuspended in lysis buffer. Protein concentrations were determined with a Bradford assay (Bio-Rad). V-ATPase western blots were quantified using Image Lab software (Bio-Rad). V-ATPase subunits in each fraction were normalized to ubiquitous lysosomal proteins.

## Lysosomal pH measurement with lysosensor

For the lysosensor assay, cells were plated and left to attach overnight in a clear bottom 96-well black border plate (Corning). Next, cells were pulsed with media containing 0.2 mg/ml Lysosensor Yellow/Blue DND-160 dextran (LS) for 4 h, washed three times and chased in fresh media for 4 h. Any treatments as indicated were performed during chase period. For each experiment, a pH calibration curve was generated. To this end, cells were incubated for 10 min at 37 °C in sodium acetate-acetic acid calibration buffers (ranging from pH 4.2 to pH 5.2), and lysosomal pH clamped to the buffer pH by addition of 10 μM nigericin and 10 μM monensin. Dual-emission ratiometric measurements were performed with a Multi-Mode Plate Reader Synergy H1 (BioTek): excitation 329 nm/emission 440 nm and excitation 380 nm/emission 540 nm. Based on the emission intensity ratio 440 nm/540 nm, lysosomal pH values were calculated using the calibration curve.

## Fluorescence microscopy

Albumin uptake and albumin degradation experiments were essentially performed as described previously[8,37]. Cells were plated on 8-well chambered coverslips (IBIDI) and left to attach overnight.

For dextran uptake experiments, cells were incubated with 0.1 mg/ml Oregon Green 488 dextran (70 kDa) or 0.1 mg/ml Alexa Fluor 568 dextran (10 kDa) for 45 min. Subsequently, cells were washed three times with ice-cold PBS, fixed with 4% formaldehyde in PBS for

15 min at room temperature, stained with 10 μg/ml Hoechst in PBS for 5 min and washed with PBS.

Lysosomal proteolysis, acidification and V-ATPase assembly were investigated by live cell imaging. For proteolysis assays with lysosomally loaded DQ BSA, cells were incubated with 0.1 mg/ml DQ BSA for 4–6 h, washed two times and chased for 3 h in fresh media to allow lysosomal accumulation of DQ BSA. Acidic organelles were labelled with 1 μM cresyl violet[19] or 10 nM lysotracker red for 15 min prior to imaging. Cathepsin activity was measured by incubation with Magic Red substrates[18] according to manufacturer's instructions for 10–12 min prior to imaging.

Lysosomal acidity was assessed by fluorescence quenching of FITC-dextran (10 kDa). Lysosomes were loaded by overnight incubation with 1 mg/ml FITC dextran, followed by 4 h chase in fresh media. V-ATPase activity was measured by quantifying the rate of FITC fluorescence quenching through acidification of neutralized lysosomes upon acute relief of V-ATPase inhibition[47]. Lysosomes were loaded with FITC-dextran (10 kDa) as above. In the last 30 min of the chase period, lysosomal pH was neutralized with 20 nM bafilomycin A1. Immediately prior to imaging, bafilomycin A1 was removed by a wash with media containing 5 mg/ml fatty acid-free BSA, and quenching of FITC fluorescence then quantified over time.

Live cell imaging was performed in a humidified chamber at 37 °C and 5% $CO_2$. Torin 1 treatment and amino acid restimulation were performed at the microscope immediately prior to imaging, if not otherwise indicated. 0.5 μg/ml Hoechst was added 30 min prior imaging. All microscopy experiments were imaged with a Leica TCS SP5 confocal microscope using a ×40 or ×63, 1.40 oil objective.

## Image analysis

Fluorescence was quantified using the particle analyser function of Fiji[48] in randomly chosen fields of view across the entirety of each sample. Mean cellular fluorescence was determined by normalizing the integrated signal density of the respective fluorescent probe to cell number. Mander's Correlation Coefficients were calculated using the Fiji plugin Coloc 2. Where indicated, individual data points in superplots were normalized to the mean of the respective control group.

To analyse subcellular changes of fluorescent probes, lysosomes were labelled by expression of Lamp1-mNeonGreen or Voa3-mScarlet as indicated. Of note, both proteins display a high degree of colocalization regardless the status of mTORC1 activity. To identify the subcellular region of interest (ROI) corresponding to Lamp1-mNeonGreen or Voa3-mScarlet-positive lysosomes, images were processed with Fiji as follows: background signal was subtracted, a Gaussian Blur function applied to robustly demarcate the lysosomal area including the organelle lumen, and ROIs were identified using the threshold and particle analyser functions of Fiji. For DQ BSA, Cathepsin Magic Red and V1B2-mNeonGreen, background signal was substracted, and integrated signal density quantified in lysosomal ROIs as defined by Lamp1-mNeonGreen or Voa3-mScarlet. For V1B2-mNeonGreen, to specifically quantify the organellar population, background subtraction was also used to decrease the diffuse cytosolic signal.

Subsequently, to quantify lysosomal DQ BSA fluorescence dequenching, DQ BSA integrated density was then normalized to the total area of Lamp1-mNeonGreen. To quantify movement of V1B2-mNeonGreen to Voa3-mScarlet-positive lysosomes, the V1B2-mNeonGreen integrated density was normalized to Voa3-mScarlet area for individual particles. For quantification of the fraction of lysosomes with high proteolytic activity (assessed by fluorescence dequenching of DQ BSA or Cathepsin Magic Red), the Fiji auto-threshold function was used in torin 1-treated cell, which display high lysosomal catabolic activity. The average auto-threshold value was then applied to all conditions of the same experiment, and the area of DQ BSA or Cathepsin Magic Red signal expressed as a percentage of

the total lysosomal area as defined by Lamp1-mNeonGreen was plotted.

## Sample preparation for proteomics

Magnetic enrichment of lysosomes was essentially performed as described previously[22]. Lysosomes were loaded with ferromagnetic nanoparticles (DexoMAG C, Liquids Research) by incubating MEFs for 14 h in medium supplemented with 10% DexoMAG and 10 mM HEPES. Cells were rinsed twice with ice-cold PBS. Next, cells were scraped into wash buffer (10 mM Triethanolamine, 250 mM sucrose, 10 mM acetic acid, protease inhibitors and phosphatase inhibitors) and pelleted by centrifugation at $120 \times g$ for 5 min. Cells were resuspended in homogenization buffer (wash buffer + 1 mM EDTA) and homogenized by 10 strokes with a KIMBLE Dounce tissue grinder with a large clearance pestle (Sigma-Aldrich). To obtain the post-nuclear supernatant (PNS) fraction, nuclei were pelleted by centrifugation at $200 \times g$ for 10 min. To obtain the enriched lysosomal fraction, the PNS was loaded on an LS MACS column (Miltenyi Biotec) equilibrated with 1 ml 0.5% BSA in PBS, washed twice with 1 ml column wash buffer (0.1 mM sucrose in PBS, protease and phosphatase inhibitors), and eluted with 400 µl elution buffer (PBS, 0.5 mM sucrose, protease and phosphatase inhibitors). Samples for biological replicates were independently generated on 5 consecutive days. Protein concentrations were determined by Bradford assay and equal protein amounts subjected to further analysis. Starting from two 15 cm plates of MEFs (80% confluence), the magnetic isolation yielded 500–600 µg protein in the enriched lysosomal fraction.

Lysosomal integrity of the enriched lysosomal fraction was performed as described[49]. Cathepsin B activity was measured using the substrate Z-Phe-Arg-7-amido-4-methylcoumarin hydrochloride (Z-Arg-Arg-AMC) (Santa Cruz sc-3136) in lysosomal fractions prepared as above in assay buffer (HBSS, bicarbonate-free, 0.6 mM $CaCl_2$, 0.6 mM $MgCl_2$, 2 mM L-cysteine, 25 mM pipes, 1 mM pefablock). As positive control, 0.1% Triton X-100 was added to release lysosomal contents. Fluorescence dequenching of Z-Arg-Arg-AMC upon hydrolysis was quantified with excitation at 355 nm and emission at 460 nm.

For SILAC-based Co-IP proteomics, the following isotopes were used: light isotopes (unlabelled Arg and Lys), medium isotopes ($^{13}C_6$-Arg and Lys-D4) and heavy isotopes ($^{13}C_6$, $^{15}N_4$-Arg and Lys-D9). Isotope-labelled amino acids were purchased from Cambridge Isotopes Laboratories or Silantes. MEFs expressing empty vector and either HA-V1B2 or Flag-Voa3 were cultured in light, medium or heavy isotope-containing media for three passages. Prior to the Co-IP, cells were placed in fresh media (full media or amino acid-free media) for 1 h, where indicated followed by amino acid restimulation for 30 min, using the respective isotope-labelled amino acids. Individual experimental conditions were labelled with either light, medium or heavy isotopes, and isotopes swapped between conditions in different experimental replicates to avoid isotope bias. Samples for biological replicates were independently generated on 4 consecutive days. After experimental treatments, MEFs were rinsed with ice-cold PBS, lysed in ice-cold Co-IP lysis buffer (40 mM HEPES pH 7.4, 150 mM $NaCl_2$, 2 mM EDTA, 2,5 $mgCl_2$, 1% Triton X-100, 5% glycerol, protease and phosphatase inhibitors) for 15 min, soluble lysate fractions isolated by centrifugation at $18,000 \times g$ for 5 min, and protein concentrations determined with the Pierce BCA Protein Assay (Thermo Fisher Scientific). Next, 5 mg of protein per experimental condition was mixed, incubated with 15 µg primary antibody (anti-HA or anti-Flag) and Dynabeads protein G (Life Technologies) for 2 h at 4 °C. The immunocomplexes were washed once with lysis buffer, released by addition of laemmli buffer (1x), and proteins finally precipitated with chloroform-methanol.

## LC-MS/MS analysis

Proteins were run for 0.5 cm into an SDS-PAGE. After Coomassie staining, the total sample was cut out and used for subsequent digestion using trypsin as described previously[50], adapted to a DigestPro MSi robotic system (INTAVIS Bioanalytical Instruments AG). Peptides were resuspended in loading buffer containing 2.5% Hexafluoro-2-propanol, 0.1% TFA in water. Analysis was carried out on an Ultimate 3000 UPLC system (Thermo Fisher Scientific) directly connected to an Orbitrap Exploris 480 (SILAC Co-IP) or a Q-Exactive HF-X (Dexomag-enriched lysosomal fractions) mass spectrometer. Analysis time and method was chosen to accompany expected sample complexity and set to 90 min for lysosomal fractions and to 120 min for the SILAC Co-IP. Prior to the analytical separation, peptides were online desalted on a trapping cartridge (Acclaim PepMap300 C18, 5 µm, 300 Å wide pore; Thermo Fisher Scientific) for 3 min using 30 µl/min flow of 0.05% TFA in water. The analytical multistep gradient was carried out on a nanoEase MZ Peptide analytical column (300 Å, 1.7 µm, 75 µm × 200 mm, Waters) using solvent A (0.1% formic acid in water) and solvent B (0.1% formic acid in acetonitrile). The concentration of B was linearly ramped from 2 to 30% for a varying time with respect to the total analysis procedure (72 min or 102 min), followed by a quick ramp to 78% B; after two minutes the concentration of B was lowered back to 2% and a 10 min equilibration step appended. Eluting peptides were analysed in the mass spectrometer using data-dependent acquisition (DDA) mode. A full scan (OE480: 60,000 resolution, 380–1400 $m/z$, 300% AGC target, 45 ms maxIT; HF-X: 120,000 resolution, 375–1500 $m/z$, 3e6 AGC target, 54 ms maxIT) was followed by 2 s (OE480) or up to 18 s (HF-X) MS/MS scans. Peptide features were isolated with a window of 1.4 $m/z$ (OE480) or 1.6 $m/z$ (HF-X) and fragmented using 26% NCE (OE480) or 27% (HF-X). Fragment spectra were recorded at 15k resolution (OE480: 100% AGC target, 54 ms maxIT; HF-X: 1e5 AGC target and 22 ms maxIT). Unassigned and singly charged eluting features were excluded from fragmentation.

## Proteomics data analysis

Data analysis was carried out by MaxQuant[51] using an organism-specific database extracted from UniProt (UP000000589_10090.fasta; download 2020-02-26; number of entries 55435) under default settings. Identification FDR cut-offs were 0.01 on peptide level and 0.01 on protein level. The match between runs option was enabled to transfer peptide identifications across RAW files based on accurate retention time and m/z. LFQ quantification was done using a label-free quantification approach based on the MaxLFQ algorithm[52]. A minimum of 2 quantified peptides per protein was required for protein quantification. SILAC quantification was done using triplex approach with medium channel Arg6, Lys4 and heavy channel Arg10, Lys9 and unlabelled amino acids as light channel. A minimum of 2 quantified peptides per protein was required for protein quantification, Re-quantify option was enabled to stabilize very large or small ratios[53].

## Proteomics statistical analysis

The statistical analysis was performed with the R-package limma[54]. SILAC ratios as well as LFQ[52] values were normalized via quantile normalization[55]. Adapted from the Perseus recommendations[56], protein groups with a valid SILAC ratio or non-zero LFQ intensity in 70% of the samples of at least one of the conditions were used for statistics. For missing SILAC ratios, random values drawn from a narrowed (0.3 standard deviation) intensity distribution around 0 of the individual sample were imputed. For missing LFQ values, random values drawn from a downshifted (1.8 standard deviation) and narrowed (0.3 standard deviation) intensity distribution of the individual sample were imputed. The $p$-values were adjusted with the Benjamini–Hochberg method for multiple testing[57].

## Statistical analysis

For proteomics data, p-values were calculated using limma moderated t-statistic and, where indicated, adjusted with the Benjamini–Hochberg method for multiple testing. For other experiments, p-values were calculated using a one-sample t-test with a hypothetical mean of 1 or a two-tailed unpaired t-test with Welch correction as indicated in the figure legends.

## Software

Statistical analyses were performed with GraphPad Prism version 9.0.0 (471). Microscopy images were acquired with Leica LAS AF software (v2.6.3.8173) and analysed with ImageJ/Fiji (v1.52n). Western blots were quantified using Bio-Rad Image Lab (v3.0.1.14). Cell sorting was performed using BD FACSDiva software (v8.0, FACSAria I). Proteomics data analysis was performed with MaxQuant (v1.6.14.0) and the R-package limma (v3.44.3).

## Reporting summary

Further information on research design is available in the Nature Research Reporting Summary linked to this article.

## Data availability

The proteomics data generated in this study have been deposited at ProteomeXchange via PRIDE with identifiers PXD030172, PXD030174. Other data generated in this study are provided in Supplementary Data files 1 and 2, and in the Supplementary Information and Source Data files. Source data are provided with this paper.

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

## Acknowledgements

We are grateful to Aurelio Teleman, Fabricio Loayza-Puch and all members of the Palm laboratory for experimental advice, critical discussions and comments on the manuscript. We thank Michael Knop, Marius Lemberg and Johannes Zuber for sharing reagents, and Franziska Hoffmann for technical assistance. This work was funded in part by a research grant of the German Research Foundation (DFG) to W.P. (PA 3679/2-1).

## Author contributions

E.R. and W.P. conceived the project, designed research, analysed and interpreted results, with contributions from S.R.C. and N.S.S. E.R. conducted experiments, with contributions from S.R.C., N.S.S. and M.W. M.S. and D.H. conducted and analysed proteomics experiments. W.P. supervised the study and wrote the manuscript with input from all authors.

## Funding

## Competing interests

The authors declare no competing interests.
