## [Peer Review File · Nature Communications]

Direct control of lysosomal catabolic activity by mTORC1 through regulation of V-ATPase assemblyREVIEWER COMMENTS

Reviewer #1 (Remarks to the Author):

The manuscript by Ratto E. et. al. report that mTORC1 regulates the lysosomal degradation of proteins through modulating the assembly of V-ATPase. Previous study has revealed that mTORC1 suppresses the recycling of proteins from the environment. However, the detailed mechanism was not clear. In this manuscript, the authors showed that mTORC1 could control the pH changes of lysosome by reversibly regulating the assembly of V-ATPases. This finding represents a big step in the field. However, some results in the current manuscript contradict with other studies and were not fully discussed in the manuscript.

Major points:

1. There are a few members in cathepsin family. Are they equally important for protein degradation in lysosomes? How does pH regulate the activity of Cathepsin? Earlier study has showed that cathepsin B displays the highest activity at pH 4.5 (JBC, 2001, 276:944-951) while in this manuscript the authors showed that lysosomal protease activities are higher at pH 4.3 than at pH 4.5 upon Torin treatment in Figure 2e. The authors need to address this discrepancy.
2. Consistent with the current finding, an early study (JBC, 2015, 290:27360-69) showed that amino acid depletion promotes the assembly of V-ATPase. However, in that paper, authors showed that the assembly of V-ATPase is independent of mTORC1 activity, which contradicts with the main finding in this manuscript. Authors should openly discuss this controversial problem in the manuscript.
3. In figure 3d, the changes of V1A might come from sample loading because the Vod1 and the control protein of CtsB also proportionally decreases in the presence of amino acid.

Minor:

V-ATPase has a Vo domain not a V0 domain. The author should switch from the number of

“0” to the letter of “o”.

Reviewer #2 (Remarks to the Author):

This manuscript reports experiments that support a direct role for mTORC1 in regulating lysosomal V-ATPase assembly, lysosomal pH and lysosomal catabolic activity in mammalian cells, adding to our understanding of the coordination of lysosomal function with nutrient availability. Although it had previously been reported that amino acid availability modulates V-ATPase assembly, the lack of effect of a low concentration of the mTORC1 inhibitor rapamycin led to the view that amino-acid dependent changes in V-ATPase assembly are controlled independently of mTORC1 (doi: 10.1074/jbc.M115.659128), despite mTORC1 activity being required for V-ATPase assembly in other circumstances e.g. during dendritic cell maturation (doi: 10.1074/jbc.M113.524561). Using the inhibitor torin1, the present authors provide compelling evidence that mTORC1 inhibition results in increased degradation of endocytosed protein as a consequence of increased V-ATPase assembly and decreased lysosomal pH. They also provide experimental evidence that when mTORC1 is active, V1, the peripheral sub-complex of the V-ATPase resides in the cytosol in association with the chaperonin TRiC, suggesting a mechanism to maintain a stable cytosolic pool of assembly-competent V1 sub-complexes. There are a number of matters that the authors should address, especially where experimental data needs strengthening and/or where additional literature should be cited:

Major points

1. Fig.1 and other figures/suppl figures. Many of the microscopy assays shown in the figures provide statistical analysis of several fields of view with a total of over 200 cells but from a single experiment. Whilst I understand that performing independent experiments is time consuming, for cell biology one would normally expect at least 3 independent experiments for each assay with p values based on the difference between population means – see e.g. <http://www.jcb.org/cgi/doi/10.1083/jcb.200611141>. Super-plots then provide a means of showing summary statistics on graphical representations of the entire cell-level data set (<https://doi.org/10.1083/jcb.202001064>). On some occasions it may be appropriate to show a single representative experiment, especially in supplementary figures. Whenever single experiments are being shown, it should at least be made clear that they are single representative experiments.
2. Line 160 onwards. This section needs re-writing to remove ambiguity and provide some additional data. The authors describe generating organelle-specific proteomes following amino acid starvation and re-feeding using a ‘highly enriched’ lysosome preparation prepared by magnetic isolation. The authors should give quantitative information about lysosome yield in their lysosome preparation and fold enrichment relative to non-lysosomal markers, not simply state ‘highly enriched,’ as in Line 178. Their proteomics data are in agreement with previous data on the association of V1 and Vo following amino acid depletion/re-feeding (see line 189 for refs 4 and 5, to which ref doi: 10.1074/jbc.M115.659128

should be added). On line 191 they corroborate their findings with blots of V-ATPase sub-units (Suppl Fig 3g) in what they describe as dextran-loaded lysosomes isolated by centrifugation. The Methods section in line 485 implies this is simply a 100,000g pellet. It seems irrelevant that it contains dextran-loaded lysosomes. This needs clarification. It seems very odd to obtain data from mass spec proteomics on a highly enriched fraction and then 'corroborate' with blots on a crude membrane fraction (i.e. essentially the experiment previously reported in doi: 10.1074/jbc.M115.659128). Given the focus on the use of torin in the first two figures it is also important that the authors provide mass spec proteomics data for their magnetically isolated lysosomes \pm torin, with a figure panel equivalent to that in Fig3c \pm amino acids.

3. Line 240. The co-IP of TRiC subunits with the ubiquitous V B2 isoform is not especially surprising given that coIP of TRiC subunits with the V1B1 isoform has been reported before in a kidney cell line V-ATPase interactome study(doi: 10.1152/ajpcell.00076.2010), which should be cited.

4. Line 274 onwards, TRiC ablation. This is a potentially important experiment and should not be relegated to Suppl Fig 6d. Moreover, quantification of the data in Suppl Fig 6d is essential. In addition, and as a minimum, the authors should provide supplementary data, preferably blotting and genotyping (if clones were generated and used), to confirm that TRiC ablation has resulted in loss of Cct1 or Cct2. Whilst the loss of 2 components of TRiC giving the same phenotype provides some confidence in the interpretation of the data, the authors should really attempt a rescue experiment.

5. Line 303 Discussion. The authors should re-write the Discussion to place their observations more firmly into the context of what is known about V-ATPase assembly/disassembly in mammalian cells. For example, they say nothing about the hypothesis, supported by data, that implies a role for RAB7a/RILP in V-ATPase assembly/disassembly/regulation (doi: 10.1242/jcs.175323, doi: 10.4161/cib.29616), or the evidence that Rabconnectin 3 may play a role in mammalian cell V-ATPase assembly similar to that of RAVE in yeast (reviewed in doi: 10.3389/fcell.2021.698190). Do their proteomics data sets throw any light on these matters? The authors should also refer to previous data showing that mTORC1 activity is required for V-ATPase assembly during dendritic cell maturation (doi: 10.1074/jbc.M113.524561). Mechanistically, do the authors have any suggestions as to how mTORC1 activity is required for the accumulation of cytosolic V1 or, when inhibited, results in V-ATPase assembly, beyond their proposal that the subcomplex in the cytosol is stabilised by TriC, which has been reported as a downstream effector of mTORC1. How do they square this proposal with the data showing that the effect of mTORC1 stabilizing the m6A RNA methyltransferase complex via TRiC is sensitive to rapamycin (reference 31), which has no effect on lysosomal pH ?

Minor points

1. Line 128. It would be helpful to provide a reference to rapamycin being a partial mTOR inhibitor (e.g. doi: 10.4161/auto.5.5.8504)

2. Line 178. Suppl Table 1 should also be cited alongside Fig 3b.

3. Line 204. Quantitative co-localisation data e.g. Pearson's or Manders' coefficients, should be provided for the data shown in Suppl Fig 4a.

4. Line 229. Suppl Table 2 should also be cited alongside Suppl. Fig 6b.

Reviewer #3 (Remarks to the Author):

In the manuscript, Ratto et al discover a mechanism by which cells promote lysosomal protein catabolism. They find that upon nutrient deplete condition the V-ATPase assembles at the lysosome, while upon nutrient-replete conditions V1 subunits are sequestered by the chaperone complex TRiC. The latter process reduces lysosomal acidification and thereby protein degradation. In yeast, it has previously shown that glucose metabolism can reversibly regulate V-ATPase assembly. However, the regulation of this process by mTOR in mammalian cells is an important new finding that should stimulate additional research in this area. The results are clear, and the paper is well written. A major advancement in the V-ATPase field is also that reversible assembly of the V-ATPase is studied with endogenously tagged V1 and V0 subunits in mammalian cells. I only have a few comments:

1. What happens when mTORC1 is activated, e.g. via TSC2 knockdown? Does this decrease lysosomal protein catabolism?
2. How are the initial endocytic steps affected by mTOR inhibition or activation? Some cell types have been shown to increase uptake upon mTOR activation (PMID: 16785324, PMID: 24953654). It should be discussed if this is consistent with the proposed model?
3. Does lysosomal protein catabolism occur in lysosomal or autolysosomes in nutrient-deplete conditions? Could it be that the process of autophagy boosts protein catabolism? What happens when autophagy is blocked, e.g. with 3-MA or via Atg5 inhibition? This is not at all addressed in the paper.
4. If lysosomal protein catabolism is enhanced by mTORC1 inhibition, then the resulting amino acids should immediately reactivate mTOR, which according to the model should inhibit V-ATPase activity leading to less protein catabolism. This point should be discussed.
5. What about mTOR reactivation in prolonged starvation? This is also supposed to work via amino acids resulting from autophagy. Using the doubly tagged V0/V1 subunit it would be interesting to study the dynamics of V-ATPase assembly in prolonged starvation. In this context, mTOR reactivation by amino acids could be blocked by protease inhibition.

Reviewer #4 (Remarks to the Author):

The manuscript entitled “Direct control of lysosomal catabolic activity by mTORC1 through regulation of V-ATPase assembly” identifies the mechanism by which mTORC1 suppresses the utilization of extracellular proteins as a source of amino acids, a process that was previously unknown. This is achieved when mTORC1 is active by the sequestration of V-ATPase V1 in the cytosol in association with TRiC. This results in low acidification of lysosomes, and thus shows how cells can adapt to their environment by modulating lysosomal activity. The paper thus uncover a previously uncharacterized mechanism of regulation of amino acids.

The authors convincingly demonstrate the effect of the inhibition of mTORC1 on lysosomal catabolism through an increase in lysosomal proteolysis due to lysosomal acidification. They then characterized the proteome of the endolysosomal compartment using DexoMAG magnetic beads in conditions where mTORC1 was inactivated by amino acid starvation or re-activated by amino acid restimulation. They identified the subunits of V-ATPase that are part of the V1 domain that were decreased in the lysosomal fractions of amino acid restimulated cells. This was further confirmed by fractionation, where a decrease of V1A is confirmed, while V0d1 did not display consistent changes. However, this is less convincing, considering that a decrease of V0d1 is also seen. I would suggest using other subunits to further confirm these observations (for example, V1B2 and V0a3, which were also identified by MS and are used in figure 4 and 5 could be used as well on these fractions). This would strengthen this point, and provide a logical continuation in the next figures (4 and 5).

To understand why the different domain of the V-ATPase are present in the cytosol of cells with high mTORC1 activity, they then profiled the interactome using Flag-tagged V0a3 or HA-tagged V1B2. The proteins were then quantitatively identified using SILAC-MS, in cells stimulated with amino acids or not. This identified the TRiC complex (Cct1-8) which reversibly associates with V-ATPase V1 during amino acid stimulation, providing a possible mechanism for the regulation of the activity of the V-ATPase.

This was further confirmed by removing TRiC using a CRISPR approach. Loss of Cct1 or Cct2 prevented the accumulation of cresyl violet in the lysosome in response to Torin 1 treatment. Moreover, they showed that mTORC1 inactivation induces a rapid activation of protein catabolism in lysosomes.

Overall, the authors have demonstrated that under nutrient-rich conditions, mTORC1 activation leads to elevated pH and low protease activity in the lysosomes, which is a direct consequence of the association of the V1 domain of the V-ATPase with the TRiC complex.

The authors thus conclude in a direct control of lysosomal catabolic activity by mTORC1.

One tantalizing question that remains, is how mTORC1 activity can directly regulate this association? Is the kinase activity necessary, and is the association between the TRiC complex and the V1 domain regulated directly by phosphorylation?

The authors potentially already have this data, perhaps through re-analysis of their proteomic experiments looking for modulation of S/T phosphorylation sites, and/or could use phosphor-specific

antibodies to identify the status of phosphorylation under amino acid starvation. This would provide a direct mechanism of regulation of this interaction.

Other minor suggestions:

How many repeats were performed for the SILAC experiment? What exactly is the control? Untransfected cells? Transfected cells with FLAG only? In addition, the control IP is with beads with protein-G only or beads with anti-flag antibodies? It would be important to describe this experiment properly.

Some details are missing from the methods section. In particular, which uniprot database was used? Which version of MaxQuant was used?

We thank the reviewers for their time and effort to review the manuscript, and appreciate their positive feedback and constructive suggestions. In the revised manuscript, we have addressed the reviewer's concerns as detailed below.

Reviewer #1 (Remarks to the Author):

The manuscript by Ratto E. et. al. report that mTORC1 regulates the lysosomal degradation of proteins through modulating the assembly of V-ATPase. Previous study has revealed that mTORC1 suppresses the recycling of proteins from the environment. However, the detailed mechanism was not clear. In this manuscript, the authors showed that mTORC1 could control the pH changes of lysosome by reversibly regulating the assembly of V-ATPases. This finding represents a big step in the field. However, some results in the current manuscript contradict with other studies and were not fully discussed in the manuscript.

We are grateful for the reviewer's positive assessment of our manuscript as a big step in the field. The reviewer raises several important questions regarding the consistency of our findings with previous results, which we have addressed with additional experiments and discussed in the revised manuscript as detailed below.

Major points:

1. There are a few members in cathepsin family. Are they equally important for protein degradation in lysosomes? How does pH regulate the activity of Cathepsin? Earlier study has showed that cathepsin B displays the highest activity at pH 4.5 (JBC, 2001, 276:944-951) while in this manuscript the authors showed that lysosomal protease activities are higher at pH 4.3 than at pH 4.5 upon Torin treatment in Figure 2e. The authors need to address this discrepancy.

In the original manuscript, we reported that cathepsins B and L display higher activity in the more acidic lysosomes of torin 1-treated cells, which correlated with an increase in lysosomal DQ BSA hydrolysis. Following the reviewer's suggestion, we now have directly determined the relevance of individual cathepsin family members for lysosomal protein degradation. To this end, we have used CRISPR/Cas9 to genetically ablated cathepsin B, D, L which are ubiquitously expressed and the most abundant cathepsins in our lysosomal proteome from mouse embryonic fibroblasts (Suppl. Table 1, Revised Suppl. Fig. 1c). Neither cathepsin B, D or L is required for the activation of DQ BSA degradation in response to torin 1 (Revised Suppl. Fig. 1d, e). By contrast, pharmacologically inhibiting all lysosomal proteases leads to a complete block of torin 1-induced DQ BSA degradation. Thus, cathepsins are essential for the activation of lysosomal protein degradation in response to mTORC1 inhibition, but none of the major cathepsins is individually responsible.

Cathepsins display high enzymatic activity at the acidic pH range characteristic of the lysosomal lumen (Handbook of Proteolytic Enzymes 2013, Academic Press). The regulation of cathepsin B activity by pH has been studied in some detail in *in vitro* activity assays using purified enzyme and synthetic peptide substrates. Cathepsin B contains several ionizable amino acid residues with pK_a values in the range of lysosomal pH, whose protonation status affects the rate of substrate hydrolysis (e.g. JBC 1983, 258:1650-5; Biochem J 1991, 751-7). However, although *in vitro* assays allow a detailed kinetic analysis of cathepsin activity, such experiments only approximately recapitulate the molecular environment of the lysosomal lumen and the endogenous protein substrates. This is an important caveat, because the pH-dependent activity

of cathepsin B and other cathepsin family members has been reported to vary between different assay conditions and model substrate (JBC 1983, 258:1650-5; Biochem J 1991, 751-7; Handbook of Proteolytic Enzymes 2013, Academic Press). Therefore, we are hesitant to interpret the subtle differences in pH-dependent enzyme activity that were determined in *in vitro* studies with our in-cell lysosomal proteolysis assay. In Figure 2e of the original manuscript, we show that lysosomal DQ BSA degradation is highest at a lysosomal pH of 4.3 – 4.4, which we consider to be remarkably consistent with the pH optimum of 4.5 that was determined for purified cathepsin B with an artificial substrate *in vitro*.

2. Consistent with the current finding, an early study (JBC, 2015, 290:27360-69) showed that amino acid depletion promotes the assembly of V-ATPase. However, in that paper, authors showed that the assembly of V-ATPase is independent of mTORC1 activity, which contradicts with the main finding in this manuscript. Authors should openly discuss this controversial problem in the manuscript.

The reviewer raises an important issue, the previous report by Forgac and colleagues that assembly of the mammalian V-ATPase was not regulated by mTORC1 (JBC, 2015, 290:27360-69). This conclusion was based on results that V-ATPase assembly did not increase in response to rapamycin treatment. The apparent discrepancy to our findings is explained by the property of rapamycin to act as a partial and selective mTORC1 inhibitor: Certain mTORC1 phosphorylation sites are highly sensitive to rapamycin, whereas many other mTORC1 phosphorylation sites are entirely rapamycin-insensitive. However, all mTORC1 targets are potently inhibited by mTOR kinase inhibitors such as torin 1 (e.g. JBC, 2009, 284 8023-32; Science, 2013, 341:1236566). The differences between distinct classes of mTOR inhibitors are recapitulated in our signaling assays: Suppl. Fig. 2h shows that rapamycin does not inhibit Ulk1 phospho-S757, an established rapamycin-insensitive site, and only partially inhibits phosphorylation of 4EBP1, which contains rapamycin-sensitive (S65) and insensitive (T37, T46) sites – by contrast, all of these phosphorylation events are strongly inhibited by two distinct mTOR kinase inhibitors – torin 1 and AZD8055. Therefore, our findings are in full agreement with the results from Forgac and colleagues: In the original manuscript, we showed that rapamycin does not decrease lysosomal pH (Fig. 2c). To further corroborate this, we now have performed additional experiments showing that rapamycin does not increase lysosomal V-ATPase assembly, cresyl violet accumulation and DQ BSA degradation (Revised Suppl. Fig. 2d – g, Suppl. Fig. 5a). This strikingly differs from the effects of mTOR kinase inhibitors and amino acid starvation, which potently induce lysosomal V-ATPase assembly, acidification and protein degradation. Thus, our results identify V-ATPase assembly as a rapamycin-insensitive effector process downstream of mTORC1. We clarify this critical issue in the revised manuscript.

3. In figure 3d, the changes of V1A might come from sample loading because the Vod1 and the control protein of CtsB also proportionally decreases in the presence of amino acid.

In Fig. 3d, e of the original manuscript, V1A and Vod1 were quantified in 7 independent experiments and normalized to cathepsin B. Thus, poor sample loading is unlikely to explain the significant increase of membrane-associated V1A in response to amino acid restimulation. Nevertheless, we have performed additional fractionation experiments and also included V1E1 as an additional V₁ domain subunit to improve the conclusiveness of results. The membrane-bound fractions of V1E1 and V1A significantly increase in response to torin 1 treatment, and conversely decrease in response to amino acid restimulation (Revised Fig. 3d, e).

Minor:

V-ATPase has a V_o domain not a V_0 domain. The author should switch from the number of "0" to the letter of "o".

Thanks!

Reviewer #2 (Remarks to the Author):

This manuscript reports experiments that support a direct role for mTORC1 in regulating lysosomal V-ATPase assembly, lysosomal pH and lysosomal catabolic activity in mammalian cells, adding to our understanding of the coordination of lysosomal function with nutrient availability. Although it had previously been reported that amino acid availability modulates V-ATPase assembly, the lack of effect of a low concentration of the mTORC1 inhibitor rapamycin led to the view that amino-acid dependent changes in V-ATPase assembly are controlled independently of mTORC1 (doi: 10.1074/jbc.M115.659128), despite mTORC1 activity being required for V-ATPase assembly in other circumstances e.g. during dendritic cell maturation (doi: 10.1074/jbc.M113.524561). Using the inhibitor torin1, the present authors provide compelling evidence that mTORC1 inhibition results in increased degradation of endocytosed protein as a consequence of increased V-ATPase assembly and decreased lysosomal pH. They also provide experimental evidence that when mTORC1 is active, V1, the peripheral sub-complex of the V-ATPase resides in the cytosol in association with the chaperonin TRiC, suggesting a mechanism to maintain a stable cytosolic pool of assembly-competent V1 sub-complexes. There are a number of matters that the authors should address, especially where experimental data needs strengthening and/or where additional literature should be cited:

We thank the reviewer for considering our findings to be compelling. We appreciate the constructive criticism and detailed experimental suggestions to strengthen our conclusions and better embed our findings in the framework of the published literature.

Major points

1. Fig.1 and other figures/suppl figures. Many of the microscopy assays shown in the figures provide statistical analysis of several fields of view with a total of over 200 cells but from a single experiment. Whilst I understand that performing independent experiments is time consuming, for cell biology one would normally expect at least 3 independent experiments for each assay with p values based on the difference between population means – see e.g. <http://www.jcb.org/cgi/doi/10.1083/jcb.200611141>. Super-plots then provide a means of showing summary statistics on graphical representations of the entire cell-level data set (<https://doi.org/10.1083/jcb.202001064>). On some occasions it may be appropriate to show a single representative experiment, especially in supplementary figures. Whenever single experiments are being shown, it should at least be made clear that they are single representative experiments.

We thank the reviewer for the detailed and constructive advice, which has helped us to substantially improve the quality and statistical significance of our quantitative microscopy experiments. Following the reviewer's suggestion, we have replaced all quantifications of microscopy data in Figure 1 (Fig. 1b, d, g, i, k) as well as for other key experiments with super-plots from at least three biologically independent experiments (Fig. 2b; Fig. 5h, l; Fig. 6d, g; Suppl. Fig. 1g; Suppl. Fig. 2b, e, g) or plots of mean values from independent experiments (Fig. 5j). For consistent statistical treatment of the data, all p values are now calculated based on the population means from at least three independent experiments. In addition, figure legends now specify whether data depict values from several independent experiments or one single representative experiment.

2. Line 160 onwards. This section needs re-writing to remove ambiguity and provide some additional data. The authors describe generating organelle-specific proteomes following amino acid starvation and re-feeding using a 'highly enriched' lysosome preparation prepared by magnetic isolation. The authors should give quantitative information about lysosome yield in their lysosome preparation and fold enrichment relative to non-lysosomal markers, not simply state 'highly enriched,' as in Line 178. Their proteomics data are in agreement with previous data on the association of V_1 and V_0 following amino acid depletion/re-feeding (see line 189 for refs 4 and 5, to which ref doi: 10.1074/jbc.M115.659128 should be added). On line 191 they corroborate their findings with blots of V-ATPase sub-units (Suppl Fig 3g) in what they describe as dextran-loaded lysosomes isolated by centrifugation. The Methods section in line 485 implies this is simply a 100,000g pellet. It seems irrelevant that it contains dextran-loaded lysosomes. This needs clarification. It seems very odd to obtain data from mass spec proteomics on a highly enriched fraction and then 'corroborate' with blots on a crude membrane fraction (i.e. essentially the experiment previously reported in doi: 10.1074/jbc.M115.659128). Given the focus on the use of torin in the first two figures it is also important that the authors provide mass spec proteomics data for their magnetically isolated lysosomes \pm torin, with a figure panel equivalent to that in Fig3c \pm amino acids.

Thanks for the detailed suggestions to improve the lysosomal proteomics part, we have revised the results and methods sections accordingly. Regarding yield and purity of the method, from two 15 cm plates of MEFs at \sim 80 % confluence we recovered 500 – 600 μ g of the enriched lysosomal fraction. The reviewer makes the valid point that quantitative information about the enrichment of lysosomal proteins relative to non-lysosomal markers would be helpful. In western blots of the same samples that were subjected to proteomics, we analyzed several abundant non-lysosomal markers, which were hardly detectable in the magnetically enriched lysosomal fraction (Fig. 3a). We are hesitant to quantify these blots, because this would very likely overestimate the enrichment factors for lysosomal over non-lysosomal proteins. Instead, we have calculated enrichment factors using the quantitative proteomics dataset: The average enrichment of lysosomal proteins / non-lysosomal proteins after magnetic enrichment (i.e. in column eluate vs. post-nuclear supernatant fractions) was $>$ 45-fold; V-ATPase and mTORC1 subunits were on average even enriched by 62-fold and 80-fold, respectively.

In the original manuscript, the key finding of Fig. 2 – a decrease in lysosomal pH in response to mTORC1 inactivation – was demonstrated both for amino acid starvation and for mTOR kinase inhibitors (Fig. 2c, d). Nevertheless, we agree with the reviewer that it would be interesting to analyze changes in the lysosomal proteome of torin 1-treated cells, and now have performed these experiments. Indeed, amino acid restimulation and torin 1 have opposite effects on V-ATPase abundance in magnetically enriched lysosomal fractions (Reviewer 2 – Figure 1). However, the effect of torin 1 was less pronounced and we consider this experiment insufficiently clear to be included in the manuscript. To corroborate the proteomics data, in the original manuscript we used two additional biochemical assays to investigate V_1 domain recruitment to membranes, which predominantly reflects changes in the lysosomal pool: pelleting of dextran-weighted lysosomes at 20,000 g, or membrane pelleting at 100,000 g. We apologize for the confusing presentation of these assays. As the reviewer points out, magnetic enrichment produces a more pure lysosomal fraction, but centrifugation of dextran-loaded lysosomes has the advantage to be faster, which in our hands retains membrane-bound V_1 domains more robustly. To improve the clarity of the results section, we have removed the ultracentrifugation data and performed additional replicates for dextran-loaded lysosomes, both for amino acid starvation-restimulation and torin 1 treatment (Revised Fig. 3d, e).

Reviewer 2 - Figure 1 | Comparison of amino acid restimulation and torin 1 on lysosomal V-ATPase levels. Changes in lysosomal proteins in MEFs after 1 h aa starvation + 30 min aa restimulation (+aa) versus 1 h aa starvation (- a) or after 1 h ± torin 1 [250 nM]. Lysosomes were magnetically enriched, proteins quantified by label-free mass spectrometry (n = 5), and proteins with log₂ fold change > 2 enrichment in eluate vs. PNS fraction analyzed. V-ATPase subunits (marked in red) display reciprocal changes in lysosomal fractions after amino acid restimulation and torin 1 treatment.

In our experience, it is challenging to detect changes in lysosomal V₁ domain recruitment by biochemical fractionation approaches. This limitation was echoed by previous observations from Forgac and colleagues that ‘It is important to note that, during cell lysis, to facilitate fractionation of membrane and cytosolic cellular components, intact pumps may be disrupted, leading to an increased amount of V₁ in the cytosolic fractions. Therefore, this method may lead to an underestimation of the absolute amount of assembled pumps *in vivo*.’ (doi: 10.1074/jbc.M115.659128). It was exactly these considerations that led us to develop a first live imaging assay for V-ATPase assembly. Live imaging reveals striking changes in V-ATPase assembly/disassembly in response to both torin 1 and amino acid starvation – restimulation, which confirms the results from biochemical experiments and provides unprecedented insights into subcellular localization and kinetics of V-ATPase assembly (Fig. 4).

3. Line 240. The co-IP of TRiC subunits with the ubiquitous V B2 isoform is not especially surprising given that coIP of TRiC subunits with the V1B1 isoform has been reported before in a kidney cell line V-ATPase interactome study(doi: 10.1152/ajpcell.00076.2010), which should be cited.

Thanks for pointing out the previous study, which identified the TRiC subunits Cct1 (Tcp1) and Cct3 in a co-IP with V1B1 (we assume the reviewer refers to doi: 10.1038/srep14827). We added this citation in the revised manuscript. We believe that a major advance of our study is the separate pull down of V₁ domain and V_o domain subcomplexes. While the whole V-ATPase complex and previously identified lysosomal interaction partners are co-IPed with either V1B2 or VoA3, all TRiC subunits (Cct1-8) specifically co-IP with V1B2, suggesting that the V₁ domain exists in two distinct pools in association with either cytosolic TRiC or membrane-integral V_o.

4. Line 274 onwards, TRiC ablation. This is a potentially important experiment and should not be relegated to Suppl Fig 6d. Moreover, quantification of the data in Suppl Fig 6d is essential. In addition, and as a minimum, the authors should provide supplementary data, preferably

blotting and genotyping (if clones were generated and used), to confirm that TRiC ablation has resulted in loss of Cct1 or Cct2. Whilst the loss of 2 components of TRiC giving the same phenotype provides some confidence in the interpretation of the data, the authors should really attempt a rescue experiment.

We thank the reviewer for highlighting the importance of Suppl. Fig. 6d and agree that these result should be included in the main figure. A caveat of the original experiment was the use of cresyl violet, an ambiguous probe that integrates both lysosomal pH and lysosomal abundance, which can change during genetic and thus inevitably slow manipulations. To address this, we now have performed additional experiments to specifically examine lysosomal acidity using fluorescence quenching of FITC-dextran, similar to the lysosomal re-acidification kinetics shown in Revised Fig. 5i, j. The results show that Cct1 and Cct2-deficient cells have less acidic lysosomes than control cells under basal conditions, and are impaired in lysosomal acidification in response to torin 1 treatment (Revised Fig. 5g, h).

We apologize for not having included western blots for Cct1 and Cct2 sgRNAs in the original manuscript, this is a key control. TRiC is an essential protein complex, and we were thus unable to generate single cell-derived Cct1 or Cct2 knockout clones. To circumvent this issue, we originally targeted either gene in a pooled cell population through stable sgRNA/Cas9 expression via lentiviral transduction. In the meantime, we have greatly improve our CRISPR/Cas9 tools by generating MEFs harboring doxycycline-inducible Cas9, which allows more reproducible and robust depletion of essential genes through time-controlled induction of CRISPR editing (doi: 10.1038/s41586-021-04035-8). The depletion of Cct1 and Cct2 in an induced pool of knockout cells is now shown in Revised Figure 5e. Following the reviewer's suggestion, we additionally have performed rescue experiments. To this end, we have generated Cct1 and Cct2 expression constructs where we introduced synonymous nucleotide changes in the sgRNA binding sites to circumvent Cas9 editing. Expression of Cas9-resistant Cct1 or Cct2 expression constructs completely rescues the decrease in V-ATPase V₁ subunits and lysosomal DQ BSA degradation that is caused by depletion of endogenous Cct1 or Cct2 (Suppl. Fig. 7e, k, l).

5. Line 303 Discussion. The authors should re-write the Discussion to place their observations more firmly into the context of what is known about V-ATPase assembly/disassembly in mammalian cells. For example, they say nothing about the hypothesis, supported by data, that implies a role for RAB7a/RILP in V-ATPase assembly/disassembly/regulation (doi: 10.1242/jcs.175323, doi: 10.4161/cib.29616), or the evidence that Rabconnectin 3 may play a role in mammalian cell V-ATPase assembly similar to that of RAVE in yeast (reviewed in doi: 10.3389/fcell.2021.698190). Do their proteomics data sets throw any light on these matters? The authors should also refer to previous data showing that mTORC1 activity is required for V-ATPase assembly during dendritic cell maturation (doi: 10.1074/jbc.M113.524561).

Thanks for the detailed reference suggestions, which have helped us to better contextualize our results. As the reviewer points out, V-ATPase assembly has been studied in mechanistic detail in yeast, but the mechanisms controlling mammalian V-ATPase assembly are less well understood. Our proteomics experiments did not detect interactions between V-ATPase subunits and RILP or Rabconnectin 3. However, we cannot exclude the possibility that these proteins interact transiently, e.g. during V-ATPase assembly, which would make it difficult to detect them in co-IP experiments. Instead, we have exploited our live imaging assay to address the relevance of putative V-ATPase assembly factors. CRISPR/Cas9-mediated deletion of the homologous DMXL1 and DMXL2 (Rabconnectin 3) blocks the increase in V1B2 and cresyl violet signal in response to torin 1 treatment (Suppl. Fig. 5d). This suggests

that DMXL1/2 are essential for mammalian V-ATPase assembly in response to mTORC1 inactivation, consistent with the well-documented function of the yeast RAVE complex in this context. In similar experiments with sgRNAs targeting RILP, we did not observe any decrease in V-ATPase assembly and lysosomal acidification (Suppl. Fig. 5d). As the reviewer points out, a potential role in V-ATPase assembly was reported in one study, which showed that RILP regulates endosomal recruitment and stability of the V1G1 subunit (doi: 10.1242/jcs.175323, doi). However, our results suggest that RILP is not required for V-ATPase assembly in response to mTORC1 inactivation. Finally, we added suggested reference showing that cluster disruption of dendritic cells in the presence of rapamycin (20 h treatment) results in decreased V-ATPase assembly (doi: 10.1074/jbc.M113.524561).

Mechanistically, do the authors have any suggestions as to how mTORC1 activity is required for the accumulation of cytosolic V1 or, when inhibited, results in V-ATPase assembly, beyond their proposal that the subcomplex in the cytosol is stabilised by TRiC, which has been reported as a downstream effector of mTORC1. How do they square this proposal with the data showing that the effect of mTORC1 stabilizing the m6A RNA methyltransferase complex via TRiC is sensitive to rapamycin (reference 31), which has no effect on lysosomal pH?

In the discussion section of the original manuscript, we proposed a model that TRiC could function as a downstream effector of mTORC1 which regulates V-ATPase assembly. This hypothesis was supported by findings of Blenis and colleagues, who identified Cct2 S260 as a residue that is phosphorylated by S6 kinase (doi: 10.1074/jbc.M900097200), as well as the more recent study by Perrimon and colleagues, who showed that rapamycin or genetic depletion of TRiC subunits decreases protein levels of the MTC complex in flies and human cells (doi: 10.1073/pnas.2021945118). We agree with the reviewer that the mechanistic connection between mTORC1 and TRiC is a critical issue, which we now have characterized in more detail with additional experiments: First, we have generated phosphorylation-deficient Cct2 S260A and phosphomimetic Cct2 S260D variants and ectopically expressed them in cells deficient for endogenous Cct2. Cells expressing Cct2 wild type, S260A or S260D displayed comparable basal levels of DQ BSA degradation and strongly increased DQ BSA degradation in response to torin 1 (Revised Suppl. Fig. 7k). Consistently, ectopic expression of Cct2 wild type, S260A and S260D rescued levels of V₁ domain subunits to a similar extent (Revised Suppl. Fig. 7l). Thus, phosphorylation of Cct2 S260 is neither required nor sufficient for the ability of mTORC1 to suppress lysosomal acidification and proteolysis. We also examined the role of S6 kinase directly, which is a rapamycin-sensitive mTORC1 effector that phosphorylates Cct2 S260. Treating cells with the S6 kinase inhibitor LY2584702 did not increase DQ BSA degradation (Revised Suppl. Fig. 7g, h).

Cct2 S260 is the only phospho-site on any TRiC subunit known to be regulated by the mTORC1 pathway. Nevertheless, we have considered the possibility that other residues on TRiC or V₁ domain subunits might be phosphorylated by mTORC1. To address this, we now have generated a phosphoproteome of cells ± torin 1 or amino acid starvation, which did not identify any further S/T phospho-sites on TRiC or V₁ domain subunits that respond to changes in mTORC1 activity (E.R., M.S., D.H., W.P., unpublished results). Consistently, PhosphoSite (www.phosphosite.org) does not report putative mTORC1 pathway-regulated S/T phospho-sites beyond Cct2 S260. Overall, these data argue against a model that mTORC1 or a downstream kinase regulates V-ATPase assembly by directly phosphorylating TRiC or the V₁ domain. Importantly, we do not consider this to contradict the study by Perrimon and colleagues (doi: 10.1073/pnas.2021945118), who propose a model that mTORC1 activates TRiC, based on results that the MTC complex is reduced by rapamycin or genetic depletion of

TRiC subunits. However, this study does not report any results that phosphorylation of Cct2 S260 (or any other TRiC residue) by the mTORC1 pathway is involved in this process. Of note, Cct2 S260 is not conserved in *Drosophila*, yet rapamycin or TRiC depletion decreases MTC in flies. Thus, whether mTORC1-S6K-mediated phosphorylation of Cct2 directly activates TRiC towards specific client proteins yet remains to be demonstrated.

Taken together, our new results suggest a refined model that the V-ATPase V₁ domain depends on TRiC for stabilization in the cytosol. When TRiC is depleted, V₁ domain subunits decrease and lysosomal acidification and proteolysis in response to mTORC1 inactivation are consequently suppressed. However, our new results strongly suggest that mTORC1 does not suppress V-ATPase assembly through direct regulation of TRiC, but rather through another effector protein, conceivably a V-ATPase assembly factor, which might indirectly promote the association of V₁ domains with TRiC by blocking their assembly with V_o domains. Experimentally testing this model and identifying the unknown mTORC1 target is an important next step. However, this will require an entirely new series of extensive biochemical and genetic experiments that are beyond the scope of the present study. We have revised the discussion accordingly.

Minor points

1. Line 128. It would be helpful to provide a reference to rapamycin being a partial mTOR inhibitor (e.g. doi: 10.4161/auto.5.5.8504)

This is a good point, we have added a reference to the paper by Sabatini and colleagues (<https://doi.org/10.1074/jbc.M900301200>) that identified rapamycin-resistant functions of mTORC1.

2. Line 178. Suppl Table 1 should also be cited alongside Fig 3b.

Thanks for the suggestion, we have added the reference to Suppl. Table 1.

3. Line 204. Quantitative co-localisation data e.g. Pearson's or Manders' coefficients, should be provided for the data shown in Suppl Fig 4a.

We have added a quantification of the co-localization of Voa3-mScarlet with Lamp1-mNeonGreen, and *vice versa* using Manders' Correlation Coefficient (Suppl. Fig. 4a – c).

4. Line 229. Suppl Table 2 should also be cited alongside Suppl. Fig 6b.

Thanks for the suggestion, we have added the reference to Suppl. Table 2.

Reviewer #3 (Remarks to the Author):

In the manuscript, Ratto et al discover a mechanism by which cells promote lysosomal protein catabolism. They find that upon nutrient deplete condition the V-ATPase assembles at the lysosome, while upon nutrient-replete conditions V1 subunits are sequestered by the chaperone complex TRiC. The latter process reduces lysosomal acidification and thereby protein degradation. In yeast, it has previously shown that glucose metabolism can reversibly regulate V-ATPase assembly. However, the regulation of this process by mTOR in mammalian cells is an important new finding that should stimulate additional research in this area. The results are clear, and the paper is well written. A major advancement in the V-ATPase field is also that reversible assembly of the V-ATPase is studied with endogenously tagged V1 and V0 subunits in mammalian cells. I only have a few comments:

We thank the reviewer for the enthusiastic feedback on our manuscript as an important new finding. We also appreciate the thoughtful comments and constructive suggestions, which we have addressed through additional experiments as detailed below.

1. What happens when mTORC1 is activated, e.g. via TSC2 knockdown? Does this decrease lysosomal protein catabolism?

Following the reviewer's suggestion, we have targeted Tsc2 with two distinct sgRNAs in MEFs harboring doxycycline-inducible Cas9. Induction of Cas9 efficiently depletes Tsc2, but this does not overtly increase mTORC1 signaling in full medium + 10 % FBS (Reviewer 3 – Figure 1a). Consistently, Tsc2 KO cells display only a slight and not significant reduction in lysosomal DQ BSA degradation, which is increased by torin 1 to the same extent as in control cells (Reviewer 3 – Figure 1b). However, Tsc2 knockout leads to sustained mTORC1 signaling in cells subjected to serum deprivation (Reviewer 3 – Figure 1a). This suggests that our standard cell culture conditions already activate mTORC1 to a high level where the signaling pathway does not further respond to acute loss of the repressive TSC complex. In this context, we think that mTORC1 activation can only be studied in a meaningful way by inactivation – reactivation, e.g. through amino acid starvation – restimulation. Unfortunately, we cannot use this approach to investigate lysosomal protein catabolism, because DQ BSA remains in the lysosome once degraded under mTORC1-inhibited conditions. However, we demonstrate that amino acid starvation – restimulation leads to an increase in lysosomal pH, V-ATPase disassembly and re-association of the V₁ domain with cytosolic TRiC, which strongly suggests that lysosomal proteolytic activity similarly responds to reactivation of mTORC1.

Reviewer 3 - Figure 1 | Tsc2 knockout does not change lysosomal albumin catabolism. a) MEFs harboring doxycycline-inducible Cas9 were infected with two different sgRNAs targeting Tsc2. Gene editing was induced by addition of doxycycline for 3 days. Subsequently, cells were placed for 16 h in 10 % or 0.1 % FBS, and mTORC1 signaling analyzed by western blot. b) Tsc2 KO MEFs (as shown in a) were incubated with DQ BSA ± torin 1

[400 nM] for 5 h. Dequenching of DQ BSA fluorescence by lysosomal degradation was quantified by microscopy. Data are represented as mean \pm SD (n = 10 fields of view; one out of 2 representative experiments).

2. How are the initial endocytic steps affected by mTOR inhibition or activation? Some cell types have been shown to increase uptake upon mTOR activation (PMID: 16785324, PMID: 24953654). It should be discussed if this is consistent with the proposed model?

How mTORC1 coordinates endocytosis and lysosomal catabolism is an intriguing question. The Neufeld and Simons labs showed that genetic activation of mTORC1 increases endocytosis, which at least in part occurred through transcriptional upregulation of the endocytic receptor, megalin. We have focused on fluid phase endocytic pathways – macropinocytosis and constitutive pinocytosis – which mediate non-selective uptake of extracellular proteins in bulk. In repeated experimental efforts, we have never seen any effects of acute mTORC1 inactivation by pharmacological inhibition or starvation on either of these endocytic pathways (Fig. 1a – d; PMID: 26144316). We now have performed additional experiments to examine the effect of Tsc2 knockout, but again do not see any changes in macropinocytosis or constitutive pinocytosis (Reviewer 3 – Figure 2). Thus, mTORC1 signaling conceivably regulates only certain endocytic pathways, e.g. expression of endocytic receptors. However, our previous work suggests a perhaps more interesting hypothesis. Ras and PI3-kinase activation downstream of growth factor receptors strongly enhances protein uptake through macropinocytosis, but in nutrient-rich conditions this also activates mTORC1, which blocks lysosomal protein degradation. Activation of Ras or PI3-kinase and concomitant inhibition of mTORC1, which e.g. happens in the context of nutrient starvation, synergizes to promote lysosomal nutrient generation by concertedly increasing macropinocytosis and depressing lysosomal catabolism (PMID: 26144316; PMID: 28973876). In the revised manuscript, we discuss evidence that endocytosis and lysosomal catabolism can be regulated independently by signaling and nutrient levels.

Reviewer 3 - Figure 2 | Tsc2 knockout does not change fluid phase endocytosis. a), b) MEFs harboring doxycycline-inducible Cas9 were infected with two different sgRNAs targeting Tsc2 (for knockout efficiency see Reviewer 3 – Figure 2). Gene editing was induced by addition of doxycycline for 3 days. Subsequently, cells were fed dextran for 30 min, fixed and imaged. Data are represented as mean \pm SEM (n = 3 independent experiments with 10 fields of view each).

3. Does lysosomal protein catabolism occur in lysosomal or autolysosomes in nutrient-deplete conditions? Could it be that the process of autophagy boosts protein catabolism? What happens when autophagy is blocked, e.g. with 3-MA or via Atg5 inhibition? This is not at all addressed in the paper.

We previously showed that knockout of the autophagy initiator kinases Ulk1/2 does not impair the upregulation of lysosomal albumin catabolism by mTORC1 inhibition (PMID: 26144316). Following the reviewer's suggestion, we now have re-examined the role of autophagy more stringently by deleting Atg5 to completely block autophagosome formation. Atg5 KO MEFs are deficient in the formation of LC3-II, but they degrade DQ BSA at the same level as control cells, both under basal conditions and in response to torin 1 (Suppl. Fig. 1h – j). To study the consequences of autophagy activation, we exploited the fact that V-ATPase assembly and lysosomal acidification is a rapamycin-insensitive mTORC1 effector pathway (Fig. 2c; Revised Suppl. Fig. 5a). This allowed us to trigger autophagy using rapamycin without changing lysosomal pH. Rapamycin does not increase lysosomal DQ BSA degradation (Revised Suppl. Fig. 2f, g). Overall, these results suggest that autophagy is neither required nor sufficient for the enhanced lysosomal degradation of endocytosed proteins in response to mTORC1 inhibition.

4. If lysosomal protein catabolism is enhanced by mTORC1 inhibition, then the resulting amino acids should immediately reactivate mTOR, which according to the model should inhibit V-ATPase activity leading to less protein catabolism. This point should be discussed.

Thanks for the suggestion, this is an excellent point! Indeed, lysosomal degradation of autophagic cargo also reactivates mTORC1, which negatively feeds back to autophagy initiation (PMID: 20526321). Along the same lines, we previously showed that lysosomal catabolism of endocytosed proteins activates mTORC1 in the complete absence of free extracellular amino acids (PMID: 26144316). In the revised discussion, we place our findings on mTORC1-mediated regulation of V-ATPase assembly in the context of these results.

5. What about mTOR reactivation in prolonged starvation? This is also supposed to work via amino acids resulting from autophagy. Using the doubly tagged V0/V1 subunit it would be interesting to study the dynamics of V-ATPase assembly in prolonged starvation. In this context, mTOR reactivation by amino acids could be blocked by protease inhibition.

We fully agree that time-resolved dynamics of V-ATPase assembly is an important issue that needs to be addressed, and have conducted preliminary experiments along the lines suggested by the reviewer. However, in our experience reactivation of mTORC1 through lysosomal nutrient generation is much more challenging to investigate than mTORC1 reactivation by amino acid stimulation, because the latter fully activates the pathway at a defined time and across the whole population of cells. The present work establishes experimental tools to study V-ATPase assembly and disassembly in response to cellular metabolic activities, but to do this adequately is a substantial effort that deserves its own project.

Reviewer #4 (Remarks to the Author):

The manuscript entitled “Direct control of lysosomal catabolic activity by mTORC1 through regulation of V-ATPase assembly” identifies the mechanism by which mTORC1 suppresses the utilization of extracellular proteins as a source of amino acids, a process that was previously unknown. This is achieved when mTORC1 is active by the sequestration of V-ATPase V1 in the cytosol in association with TRiC. This results in low acidification of lysosomes, and thus shows how cells can adapt to their environment by modulating lysosomal activity. The paper thus uncover a previously uncharacterized mechanism of regulation of amino acids.

We thank the reviewer for the positive appraisal of our findings and are grateful for the constructive and detailed comments. The reviewer raises several important issues, which we have addressed as delineated in the following.

The authors convincingly demonstrate the effect of the inhibition of mTORC1 on lysosomal catabolism through an increase in lysosomal proteolysis due to lysosomal acidification. They then characterized the proteome of the endolysosomal compartment using DexoMAG magnetic beads in conditions where mTORC1 was inactivated by amino acid starvation or re-activated by amino acid restimulation. They identified the subunits of V-ATPase that are part of the V1 domain that were decreased in the lysosomal fractions of amino acid restimulated cells. This was further confirmed by fractionation, where a decrease of V1A is confirmed, while V0d1 did not display consistent changes. However, this is less convincing, considering that a decrease of V0d1 is also seen. I would suggest using other subunits to further confirm these observations (for example, V1B2 and V0a3, which were also identified by MS and are used in figure 4 and 5 could be used as well on these fractions). This would strengthen this point, and provide a logical continuation in the next figures (4 and 5).

The quantification of V1A and Vod1 was based on normalization to cathepsin B, and quantification of 7 independent experiments demonstrated significant increase in membrane-associated V1A but not Vod1 in response to amino acid restimulation. Following the reviewer’s suggestions, we now have performed additional fractionation experiments to strengthen this conclusion. As an additional V₁ domain subunit, we included V1E1, whose membrane-bound fraction significantly increases in response to torin 1 treatment, and conversely decreases in response to amino acid restimulation, similar to V1A (Revised Fig. 3d, e). We agree with the reviewer that it would be elegant to include data for V1B2 and V0a3, but unfortunately did not succeed in validating antibodies that are suitable for quantitative western blotting. While we could have performed the experiments with affinity-tagged variants, an important rationale for the biochemical fractionation experiments was to detect endogenous, unaltered V-ATPase subunits to provide orthogonal validation for the live imaging assay. However, we examined V1A both in biochemical fractionation experiments and in live imaging (Suppl. Fig. 5b), confirming key results with both approaches.

To understand why the different domain of the V-ATPase are present in the cytosol of cells with high mTORC1 activity, they then profiled the interactome using Flag-tagged V0a3 or HA-tagged V1B2. The proteins were then quantitatively identified using SILAC-MS, in cells stimulated with amino acids or not. This identified the TRiC complex (Cct1-8) which reversibly associates with V-ATPase V1 during amino acid stimulation, providing a possible mechanism for the regulation of the activity of the V-ATPase. This was further confirmed by removing TRiC using a CRISPR approach. Loss of Cct1 or Cct2 prevented the accumulation of cresyl violet in the lysosome in response to Torin 1 treatment. Moreover, they showed that mTORC1

inactivation induces a rapid activation of protein catabolism in lysosomes. Overall, the authors have demonstrated that under nutrient-rich conditions, mTORC1 activation leads to elevated pH and low protease activity in the lysosomes, which is a direct consequence of the association of the V1 domain of the V-ATPase with the TRiC complex. The authors thus conclude in a direct control of lysosomal catabolic activity by mTORC1. One tantalizing question that remains, is how mTORC1 activity can directly regulate this association? Is the kinase activity necessary, and is the association between the TRiC complex and the V1 domain regulated directly by phosphorylation? The authors potentially already have this data, perhaps through re-analysis of their proteomic experiments looking for modulation of S/T phosphorylation sites, and/or could use phosphor-specific antibodies to identify the status of phosphorylation under amino acid starvation. This would provide a direct mechanism of regulation of this interaction.

Whether mTORC1 phosphorylates TRiC directly to regulate the interaction with the V-ATPase V₁ domain is an important question that was not resolved in our original manuscript. In the discussion, we proposed a model that TRiC could function as a downstream effector of mTORC1 which regulates V-ATPase assembly. This hypothesis was supported by findings of Blenis and colleagues, who identified Cct2 S260 as a residue that is phosphorylated by the mTORC1 effector, S6 kinase (doi: 10.1074/jbc.M900097200). We now have examined the relevance of S6 kinase-mediated phosphorylation of Cct2 in detail: First, we have generated phosphorylation-deficient Cct2 S260A and phosphomimetic Cct2 S260D variants and ectopically expressed them in cells where endogenous Cct2 was deleted by CRISPR/Cas9. Cct2 S260A does not increase basal levels of DQ BSA degradation, and does not abrogate the strong increase of DQ BSA degradation in response to torin 1. Similarly, Cct2 S260D does not change DQ BSA under either basal conditions or upon torin 1 treatment (Revised Suppl. Fig. 7k). Consistently, ectopic expression of Cct2 wild type, S260A and S260D rescue levels of V₁ domain subunits to a similar extent (Revised Suppl. Fig. 7l). Thus, phosphorylation of Cct2 S260 by S6 kinase (or another mTORC1 pathway component) is neither required nor sufficient for the suppression of lysosomal acidification and proteolysis by mTORC1. We also examined the role of S6 kinase directly. Treating cells with the S6 kinase inhibitor LY2584702 does not increase DQ BSA degradation and lysosomal acidification (Revised Suppl. Fig. 7g – i). Thus, mTORC1 regulates lysosomal acidification and proteolysis through a mechanism that does not involve S6 kinase-mediated regulation of TRiC.

Cct2 S260 is the only phospho-site on TRiC that is known to be regulated by the mTORC1 pathway. Nevertheless, we have considered the possibility that other residues on TRiC subunits or the V-ATPase V₁ domain might be phosphorylated by mTORC1. To test this, we now have generated a phosphoproteome of cells ± torin 1 or amino acid starvation (n = 5 for each condition). We robustly quantify multiple S/T phospho-sites on various subunits of TRiC and the V₁ domain, but none of these responds to mTORC1 inhibition or activation (E.R., M.S., D.H., W.P., unpublished results). Consistently, PhosphoSite (www.phosphosite.org) does not report any mTORC1 pathway-regulated S/T phospho-sites beyond Cct2 S260. Overall, these data argue against a model that mTORC1 or a downstream kinase regulates V-ATPase assembly by phosphorylating TRiC or the V₁ domain. Rather, these results suggest that mTORC1 regulates another effector protein, conceivably a V-ATPase assembly factor, which might indirectly promote the association of the V₁ domain with TRiC by blocking its assembly into V-ATPase complexes. Experimentally testing this model and identifying the unknown mTORC1 target is an important next step. However, this will require an entirely new series of extensive biochemical and genetic experiments that are beyond the scope of the present study. We have revised the discussion accordingly.

Other minor suggestions:

How many repeats were performed for the SILAC experiment? What exactly is the control? Untransfected cells? Transfected cells with FLAG only? In addition, the control IP is with beads with protein-G only or beads with anti-flag antibodies? It would be important to describe this experiment properly. Some details are missing from the methods section. In particular, which uniprot database was used? Which version of MaxQuant was used?

Thanks for these suggestions! We apologize for the inadequate description of the proteomics experiments, and have revised the methods section to describe the experimental procedures in detail. The SILAC Co-IPs were conducted in 4 biological replicates, with empty vector-expressing cells as controls. Control cells and bait-expressing cells were labeled with different isotopes. IPs were performed in one single assay with pooled lysates from the different experimental groups to ensure that control and bait-expressing cells are subjected to identical IP conditions, including time, protein G beads and primary antibody. The proteomics data were analyzed with MaxQuant version 1.6.14.0 and database UP000000589_10090.fasta (download 2020-02-26; number of entries 55435).

REVIEWERS' COMMENTS

Reviewer #1 (Remarks to the Author):

The authors have fully addressed my questions and I have no further concern. I support the publication of this manuscript.

Reviewer #2 (Remarks to the Author):

The authors have addressed in detail and to my satisfaction, all the specific matters I raised when reviewing the previous version of the manuscript. Through adding additional experimental data and citing a wider literature they have been able to present an even more compelling account of the role of mTORC1 in regulating lysosomal V-ATPase assembly, pH and catabolic activity in mammalian cells.

Reviewer #3 (Remarks to the Author):

My comments have now been adequately addressed. Only for point 2, discussion of endocytic mechanisms with respective references (PMID: 16785324, PMID: 24953654) is still missing in the revised version. Please add.

Reviewer #4 (Remarks to the Author):

The authors have revised Fig.3d,e and Suppl. Fig.5b to confirm the changes that were quantified. Unfortunately, the V1B2 and V0a3 did not work, but at least the data is now more convincing. I would also not recommend overexpression.

The issue on phosphorylation, which could regulate directly TRiC, has now been more extensively addressed through mutants (Suppl. Fig. 7) and inhibitors of the S6 kinase. Phosphoproteomic experiments have also been performed which identified several sites on various subunits of TRiC, but none of them were found to respond to mTORC1 inhibition or activation. It was worth the effort, and I agree that this point, this becomes an entirely new project.

The details on the proteomics experiments are now properly detailed.

All my concerns have now been addressed, and I would recommend the manuscript for publication.